# ContextAgent: Context-Aware Proactive LLM Agents with Open-World Sensory Perceptions

**Bufang Yang**[1†] **Lilin Xu**[2†] **Liekang Zeng**[1] **Kaiwei Liu**[1] **Siyang Jiang**[1] **Wenrui Lu**[1]
**Hongkai Chen**[1] **Xiaofan Jiang**[2] **Guoliang Xing**[1] **Zhenyu Yan**[1‡]

[1]The Chinese University of Hong Kong  [2]Columbia University

{bfyang,lkzeng,lk022,syjiang,wrlu,hkchen,glxing,zyyan}@ie.cuhk.edu.hk,
lx2331@columbia.edu, jiang@ee.columbia.edu

## Abstract

Recent advances in Large Language Models (LLMs) have propelled intelligent agents from reactive responses to proactive support. While promising, existing proactive agents either rely exclusively on observations from enclosed environments (e.g., desktop UIs) with direct LLM inference or employ rule-based proactive notifications, leading to suboptimal user intent understanding and limited functionality for proactive service. In this paper, we introduce ContextAgent, the first context-aware proactive agent that incorporates extensive sensory contexts surrounding humans to enhance the proactivity of LLM agents. ContextAgent first extracts multi-dimensional contexts from massive sensory perceptions on wearables (e.g., video and audio) to understand user intentions. ContextAgent then leverages the sensory contexts and personas from historical data to predict the necessity for proactive services. When proactive assistance is needed, ContextAgent further automatically calls the necessary tools to assist users unobtrusively. To evaluate this new task, we curate ContextAgentBench, the first benchmark for evaluating context-aware proactive LLM agents, covering 1,000 samples across nine daily scenarios and twenty tools. Experiments on ContextAgentBench show that ContextAgent outperforms baselines by achieving up to 8.5% and 6.0% higher accuracy in proactive predictions and tool calling, respectively. We hope our research can inspire the development of more advanced, human-centric, proactive AI assistants. The code and dataset are publicly available at https://github.com/openaiotlab/ContextAgent.

## 1 Introduction

Large Language Model (LLM) agents are revolutionizing our daily life [16], assisting users with complex tasks such as automated web navigation [12, 56, 9], software engineering [46, 51, 36], and healthcare services [4, 43, 25]. While LLM agents are receiving growing attention and adoption, most of them still function in a *reactive paradigm*: They can initiate tasks only upon explicit user instructions and yet lack the autonomy to perceive environments and offer proactive support for users.

To further reduce reliance on instructions and alleviate human cognitive workload, proactive agents emerge, which are capable of initiating tasks without explicit user queries [55, 24, 23, 52, 50]. For example, research on proactive agents have explored coding assistance [55, 24], conversation participation [23, 44], re-asking strategies to reduce ambiguity in user instructions [52], and multi-agent cooperation scenarios [50, 39]. However, their limited ability in open-world perceptions and restricted functionality for proactive service hinders their potential as personal companions.

---

[†]Equal Contribution. [‡]Corresponding Author.

39th Conference on Neural Information Processing Systems (NeurIPS 2025).

**Environmental Perception.** When explicit user instructions are absent, environment perception is crucial for proactive LLM agents. Recent studies [24, 55] proposed proactive agents for programming assistance, while they require access to specific inputs such as computer screenshots or keyboard inputs. We argue that an ideal proactive agent should be able to perceive open-world environments in the user's daily life, utilizing wearable devices such as smart glasses and earphones. By sharing the same perception as the user, the agent can understand the user's intention and provide services automatically. Besides, the hands-free nature of these ubiquitous wearable devices aligns well with the mission of proactive agents, freeing both the user's hands and mind from additional workload.

**Functionality for Proactive Services.** Current personal assistants can deliver proactive notifications via wearables, yet remain limited by static, rule-based pipelines (e.g., alerts when rapid falling is detected [3]). Recent studies [24, 55, 44, 52] propose to build proactive agents with LLMs. However, these agents only provide direct answers during user interactions, without leveraging external tools, and remain limited to enclosed environments (e.g., desktop and keyboard inputs [24, 55]). Therefore, there remains a research gap in developing a

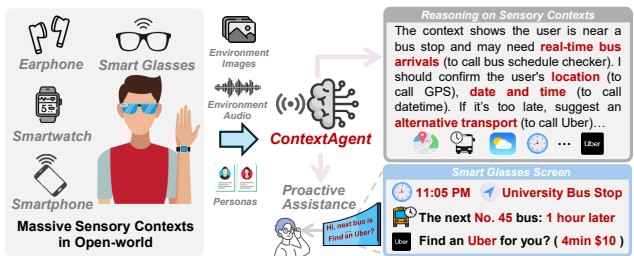

Figure 1: ContextAgent is a proactive AI assistant free of user explicit instructions. ContextAgent can continuously perceive environmental contexts (e.g., image and audio) to detect the necessity of proactive services, and provide tool-augmented assistance based on LLM reasoning.

context-aware proactive LLM agent that can exploit extensive sensory contexts to comprehensively understand user intentions, predict the necessity of proactive services, and automatically integrate external tools to deliver unobtrusive services as a personal companion.

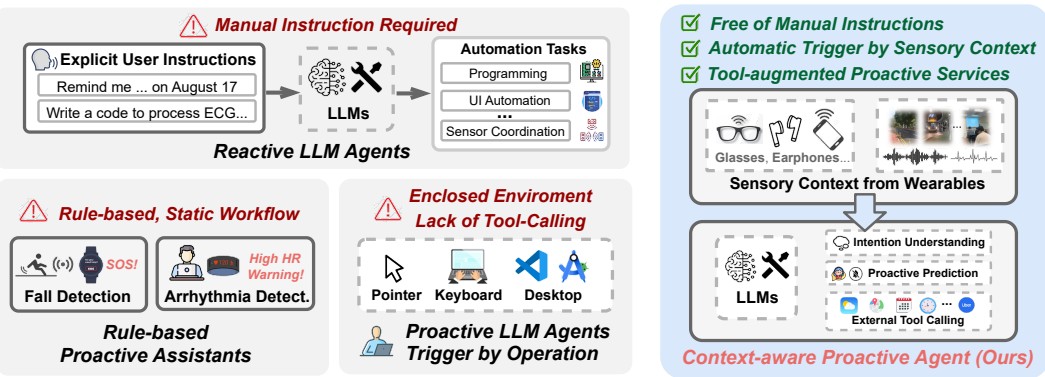

Figure 2: Comparison with existing works. Reactive LLM agents require explicit user instructions to initiate tasks. Prior proactive LLM agents focus on perceiving enclosed environments (e.g., desktop UIs) and may still require user operations (e.g., keyboard inputs) alongside direct LLM inference. In contrast, ContextAgent requires no manual instructions, harnesses massive sensory contexts from the open world, and employs LLM reasoning for tool-augmented proactive services.

In this paper, we introduce ContextAgent, the first context-aware proactive LLM agent that harnesses extensive sensory contexts for enhanced proactive services. ContextAgent first employs a proactive-oriented context extraction approach to derive both sensory and persona contexts from massive sensory perceptions such as egocentric videos and audio. We then develop a context-aware reasoner that integrates both sensory and persona contexts for reasoning, predicts the necessity of proactive services, and calls external tools when necessary. This reasoner is fine-tuned with reasoning traces distilled from advanced reasoning LLMs, enabling it to think before acting. Fig. 1 shows an example scenario where the user arrives at a bus stop just after the bus has left. ContextAgent can leverage this sensory context to proactively deliver useful services, such as real-time bus schedules, and determine whether alternative transportation is needed. By harnessing sensor perceptions from hands-free, egocentric wearables (e.g., smart glasses and earphones), along with LLM reasoning, ContextAgent moves closer toward a more ubiquitous and proactive AI assistant.

To better examine ContextAgent, we further introduce a new benchmark, ContextAgentBench, for the comprehensive evaluation of context-aware proactive LLM agents. ContextAgentBench contains 1,000 samples covering nine daily life scenarios, such as working and chitchat, and includes twenty external tools. We conduct comprehensive evaluations, comparing ContextAgent to six baselines and testing on 13 LLMs. Results show that ContextAgent achieves the state-of-the-art performance in proactive predictions and tool calling. We summarize the contributions of this work as follows.

- We raise the research problem of context-aware proactive agents that harness extensive sensory contexts surrounding humans to enhance the proactivity of the LLM agents and deliver tool-augmented proactive services, moving toward the vision of proactive personal assistants.

- We propose ContextAgent, the first framework for context-aware proactive LLM agents. ContextAgent employs a novel context extraction method that derives sensory and persona contexts from massive sensor perceptions. Additionally, we design a context-aware reasoner with think before action capabilities that can integrate both sensory and persona contexts for reasoning, predict the necessity of proactive services, and call external tools when necessary to assist the user.

- We introduce ContextAgentBench, the first benchmark for the comprehensive evaluation of context-aware proactive LLM agents. Extensive evaluation on ContextAgentBench shows that ContextAgent outperforms six baselines by achieving up to 8.5% higher accuracy for proactive predictions, 7.0% higher F1-score for tool calling, and 6.0% higher accuracy for tool arguments.

## 2    Related Works

**Reactive LLM-based Agents.** Recent studies have proposed various LLM agents to perform complex tasks, such as automated web navigation [12, 56], software engineering [46], personal assistant [44, 42], and household robotics [6]. Additionally, prior research has primarily focused on enhancing the core capabilities of LLM agents, including task planning [47], function calling [29, 21, 32], experience reflection [54, 33], generalization abilities [38, 27], and multi-agent collaboration [34, 22, 53]. Other studies have explored the LLM agents in mobile systems, such as autonomous UI operations on smartphones [37, 18, 49] and embedded programming [31, 13, 45]. However, although numerous frameworks and optimizations have been proposed, prior research has primarily focused on reactive LLM agents that require explicit textual instructions from users and cannot utilize the extensive contextual information from sensor perceptions on wearable devices to enable proactive assistance.

**Proactive LLM Agents.** Proactive agents aim to autonomously initiate services based on environmental observations, without requiring explicit user instructions, evolving from early rule-based or periodic triggers [3] to recently proposed LLM-based approaches [55, 24]. Ask-before-plan [52] employs re-asking strategies to proactively reduce ambiguity in a user's instructions and enhance subsequent planning, although it still requires an initial user query. ProAgent [50] is a proactive cooperation framework among multiple robot agents, while its proactive design primarily focuses on predicting teammates' actions in multi-agent systems rather than the user's intention. Recent studies, such as Proactive Agent [24] and CodingGenie [55], also propose proactive LLM agents that monitor the user interface environment on computer systems and proactively assist with tasks such as coding and writing. However, existing work either leverages observations on computer interfaces or employs a re-asking strategy to gather more information, without utilizing the rich sensory contexts to proactively initiate services. Moreover, prior works primarily use LLMs for direct inference rather than integrating external tools, resulting in limited proactive service functionality.

**LLM Agent Benchmark.** A diverse and large-scale benchmark is essential for the comprehensive evaluation of LLM agents. However, existing benchmarks primarily focus on reactive LLM agents [46, 29, 26, 11, 21], where the agent needs to take user instructions as inputs and perform task planning and tool calling. Although a recent work [24] proposes ProactiveBench, it is limited to an enclosed environment, i.e., desktop UI, and does not leverage the rich contextual information from multi-modal sensors on wearable devices. Additionally, ProactiveBench relies on direct LLM inference for responses, instead of calling diverse external tools. Therefore, a research gap remains in developing a comprehensive benchmark for evaluating proactive LLM agents that incorporate the rich contextual information from wearable devices for proactive reasoning with tool-calling capabilities.

# 3 Context-aware Proactive Agent Task

## 3.1 Task Definition

In contrast with existing reactive LLM agents and proactive agents that rely solely on observations from desktop interfaces or direct inference, we formalize context-aware proactive LLM agents as: $(\mathcal{T}, \mathcal{P}_\mathcal{S}, \mathcal{T}_\mathcal{C}, \mathcal{R}) = \mathcal{A}(\mathcal{S}, \mathcal{P})$, where $\mathcal{A}$ is the LLM agent, which integrates the sensory perceptions $\mathcal{S}$ and persona context $\mathcal{P}$ as input. Here $\mathcal{S}$ contains sensor perceptions from multi-modal wearables such as smart glasses and earphones, including egocentric video $\mathcal{S}_\mathcal{V}$, audio $\mathcal{S}_\mathcal{A}$, and smartphone notification $\mathcal{N}$. We denote the sensory context $\mathcal{C}$ as the implicit cues within the raw sensory perceptions $\mathcal{S}$ that help determine the need for proactive services. We also formalize that the agent should consider user personas $\mathcal{P}$ for proactive reasoning, including a person's identity, preferences, and historical behaviors. Using these contexts, the agent generates $(\mathcal{T}, \mathcal{P}_\mathcal{S}, \mathcal{T}_\mathcal{C}, \mathcal{R})$, where $\mathcal{T}$ denotes the explicit thought traces. $\mathcal{P}_\mathcal{S}$ denotes the proactive score, which triggers proactive services when $\mathcal{P}_\mathcal{S} \geq \theta$. Here, $\theta$ denotes the threshold for initiating proactive services and is a user-adjustable parameter reflecting the user's sensitivity to such services. $\mathcal{T}_\mathcal{C}$ is the planned tool chains that LLM agents should call in sequence, where $\mathcal{T}_\mathcal{C} = (t_i, a_i)_{i=1}^N, t_i \in \mathbf{T}$, with $t_i$ as each tool to be called and $a_i$ as the corresponding arguments. $\mathbf{T}$ is the tool set that the agent can use. $\mathcal{R}$ is the agent's final response, summarizing the sensory context, persona context, reasoning traces, and tool results. Note that proactive assistance is only initiated when $\mathcal{P}_\mathcal{S} \geq \theta$, otherwise the agent does not disturb the user.

## 3.2 Task Construction

Recognizing the shortcomings of existing LLM agent benchmarks, we present ContextAgentBench, the first benchmark designed to evaluate context-aware proactive LLM agents.

**Design Choices.** Our dataset includes the following key features: 1) *Sensory Context.* Our dataset contains sensory context obtained from wearables (e.g., smart glasses and earphones), which capture shared perceptions of the user *ubiquitously*. This hands-free captured sensory context is more suitable for proactive agents as it can reduce the user's physical and cognitive workload, aligning with the mission of proactive agents. 2). *Persona Context.* We incorporate diverse personas to support more comprehensive and personalized scenarios for proactive services. 3) *Proactive Assistance with Tool Calling.* The dataset targets tool-using LLM agents that map the contexts to proactive assistance by utilizing multiple external tools to generate more informative responses, rather than direct inference.

**Formulation and Exemplar Design.** Each sample in our dataset contains seven parts: $(\mathcal{S}, \mathcal{C}, \mathcal{P}, \mathcal{T}, \mathcal{P}_\mathcal{S}, \mathcal{T}_\mathcal{C}, \mathcal{R})$. Next, we introduce the design of initial exemplars.

*Multi-dimensional Context Information.* Annotators first write textual descriptions of their egocentric perceptions, including what they see, hear, and any mobile device notifications, for both proactive and non-proactive scenarios that they encounter in daily life. This sensory perception can be captured from an egocentric perspective using various wearable devices. The context information contains the visual context $\mathcal{C}_\mathcal{V}$, the acoustic context $\mathcal{C}_\mathcal{A}$, and the notifications on the smartphone $\mathcal{N}$. Annotators also summarize them into contextual information $\mathcal{C}$, providing a comprehensive description of the user's current conditions. Annotators write the user personas $\mathcal{P}$ for the sample if necessary. The persona can include any information about a person's preferences or identity.

*Proactive Score with Planned Tool Chains.* Next, annotators are instructed to analyze the current context and assign a proactive score $\mathcal{P}_\mathcal{S}$. We define $\mathcal{P}_\mathcal{S}$ on a scale from 1 to 5, where 1 means that no proactivity is required and 5 means a high level of proactivity. Annotators also receive a tool set $T$ that includes the usable tools, tool names, tool descriptions, arguments, and formats predefined by the developers. Details are in the Appendix D. For samples identified as requiring proactivity, we request annotators to further label the planned tool chains $\mathcal{T}_\mathcal{C}$, specifying the external tools that agents should use. If $\mathcal{P}_\mathcal{S} = 1$ or 2, both $\mathcal{T}_\mathcal{C}$ and $\mathcal{R}$ are None, as there is no need for proactivity.

We instruct the annotators to create samples spanning nine everyday scenarios, ranging from work to chitchat. We ask annotators to document their thought processes, including their analysis of the current context, their rationale for assigning the proactive score, and the planned tool chains. Each annotator also cross-reviewed the samples produced by others, evaluating both the format and plausibility to avoid overproactivity and ensure the correctness of annotations. Through this process, we acquire 200 human-created exemplars to serve as the seed dataset.

**Automated Diversification Pipeline.** Relying solely on manual efforts to scale the dataset presents challenges, as scenarios and contextual information created by humans may lack diversity and generalizability. Moreover, human fatigue during annotation can introduce bias, potentially compromising the dataset's quality. Therefore, we develop an automated diversification pipeline to use LLMs for data generation, producing a large-scale dataset with diverse samples.

*Information Source.* We first prepare several resources to help LLMs generate synthetic data, including the tool set (Appendix D), an extensive persona pool, and the initial exemplars. The personas in our pool are sourced from [14], which includes one billion individual identities and preferences.

*Generation with Verification.* Next, we prompt LLMs to generate diverse samples by utilizing the initial exemplars, tool set, and persona pool for reference. We employ two strategies during generation: scenario-aware and proactive score-aware. In the first strategy, we group the seed dataset by scenarios and instruct LLMs to generate samples based on specific scenarios within the nine categories. In the second strategy, LLMs are prompted to generate samples based on a specific proactive score. Details are in the Appendix B. After generation, annotators first evaluate the context and annotations for rationality. Next, we execute a script to verify the correctness of the data format and tool arguments. We perform several iterations of the above process to obtain ContextAgentBench.

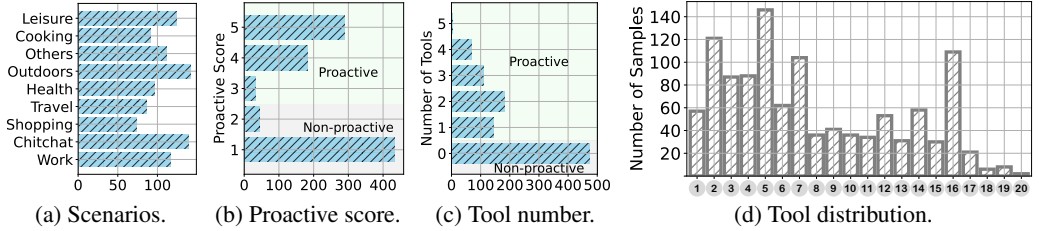

(a) Scenarios.  (b) Proactive score.  (c) Tool number.  (d) Tool distribution.

Figure 3: Statistics of ContextAgentBench, including the sample distribution across different scenarios, proactive scores, and the number and types of tools. In subfigures (a)–(c), the x-axis shows the number of samples, whereas in (d) it denotes the tool index.

**Consideration of Sensor Data.** We also collect raw sensor data from copyright-free internet platforms [1] to pair with the textual contextual information in ContextAgentBench. Specifically, we first randomly select samples from ContextAgentBench, and scrape the videos from Pexels [1] based on the textual descriptions of the visual context information. Note that we exclusively collect videos captured from an egocentric perspective. Additionally, for samples with audio conversations, we self-collect both video and audio to align with the textual context information. Finally, we obtain the ContextAgentBench-Lite, consisting of 300 human-verified samples with raw sensor data.

**Dataset Statistics.** Fig. 3 shows the statistics of our dataset. We collected 1,000 samples for ContextAgentBench and 300 samples for ContextAgentBench-Lite. Our dataset covers 9 daily life scenarios and includes 20 tool types, with each sample potentially involving the use of up to five tools. We provide more details on the dataset and tool definitions in the Appendix C.

## 4 ContextAgent Framework

This section presents the framework of ContextAgent, introducing how it utilizes the massive sensory contexts for tool-augmented proactive LLM agent services. Fig. 4 shows the overview of ContextAgent. First, ContextAgent extracts proactive-oriented contexts from multi-modal sensory perceptions. Next, ContextAgent integrates these contexts for tool-augmented proactive services.

### 4.1 Proactive-oriented Context Extraction

Previous studies focus on extracting sensory contexts and use LLMs to summarize insights [28]. However, relying solely on these sensory contexts can lead to inferior proactive predictions. Therefore, ContextAgent employs a proactive-oriented context extraction method. In ContextAgent, contexts comprise two types: *sensory context* and *persona context*. The sensory context includes insights for the user's surroundings and actions, which are crucial for inferring user intent. The persona context encompasses user personal information, including past behaviors, preferences, and identity,

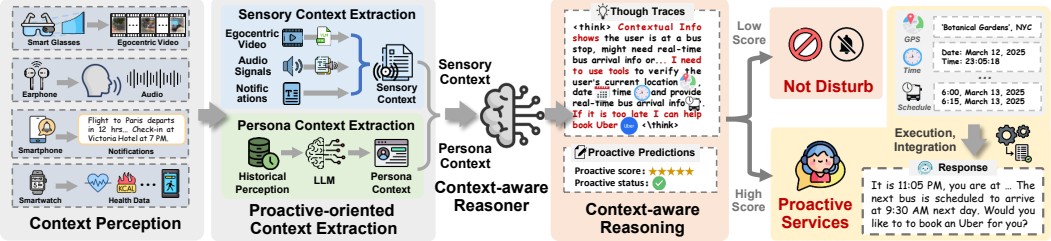

Figure 4: Overview of ContextAgent. ContextAgent extracts sensory context from massive sensor perceptions. Then it integrates both sensory and persona contexts into LLM reasoning, generating thought traces, proactive predictions, and calling external tools for proactive services when necessary.

which helps LLM agents to determine the need and urgency for proactive assistance. This subsection provides details on how ContextAgent extracts these contexts from extensive sensory perceptions.

**Sensory Context.** ContextAgent first employs Vision Language Models (VLMs) [19] to transform raw egocentric videos into visual contexts $\mathcal{C}_{\mathcal{V}}$. Although existing VLMs can generate video descriptions, they often produce overly simplistic descriptions that overlook crucial cues for understanding user intent or overly detailed, redundant insights, both of which can hinder the LLM agent's proactive predictions. Thus, instead of using zero-shot VLMs, ContextAgent employs in-context learning (ICL) to generate proactive-oriented visual contexts $\mathcal{C}_{\mathcal{V}}$. ContextAgent also employs the speech recognition model to generate audio contexts $\mathcal{C}_{\mathcal{A}}$. Details and prompts for sensory context extraction are in the Appendix B. Finally, ContextAgent integrate these contexts into the final context information $\mathcal{C} = [\mathcal{C}_{\mathcal{V}}, \mathcal{C}_{\mathcal{A}}, \mathcal{N}]$, which includes visual contexts $\mathcal{C}_{\mathcal{V}}$, audio contexts $\mathcal{C}_{\mathcal{A}}$, and textual information from smartphone notifications $\mathcal{N}$, such as calendar events and hotel reservations.

**Persona Context.** Since the need for assistance highly depends on the user's personal preferences, ContextAgent also integrates persona context into its reasoning. In this work, we use persona contexts within ContextAgentBench for experiments. In practice, these contexts can be continuously updated by utilizing LLMs to extract insights from historical sensory data like daily conversations [44].

## 4.2 Context-aware Proactive Reasoning

While existing LLM agents can handle complex tasks based on explicit user instructions [20, 56, 9], they face challenges when processing sensory contexts and correctly mapping them to the appropriate tools for proactive services. Next, we will introduce the context-aware reasoner in ContextAgent.

**Context-aware Reasoner.** ContextAgent employs a context reasoner $\mathcal{A}_{\mathcal{S}}$ to reason over the generated contexts and provide proactive services: $(\mathcal{T}, \mathcal{P}_{\mathcal{S}}, \mathcal{T}_{\mathcal{C}}) = \mathcal{A}_{\mathcal{S}}(\mathcal{C}, \mathcal{P})$. The context reasoner is an LLM that integrates both sensory context $\mathcal{C}$ and personas $\mathcal{P}$ as input to generate thought traces $\mathcal{T}$, proactive scores $\mathcal{P}_{\mathcal{S}}$, and tool chains $\mathcal{T}_{\mathcal{C}}$. We enable ContextAgent to perform think-before-act reasoning by distilling traces from advanced LLMs (e.g., Claude-3.7-Sonnet [2]) and constructing a CoT-based [35] fine-tuning dataset. During inference, once $\mathcal{P}_{\mathcal{S}} \geq \theta$, ContextAgent will initiate the proactive services. Additionally, ContextAgent generates tool chains $\mathcal{T}_{\mathcal{C}}$ for enhanced proactive services. ContextAgent will execute the planned tools sequentially and integrate their results with the sensory context, persona context, and thought traces into the LLM to generate final responses.

**Training Scheme.** We use supervised fine-tuning (SFT) with CoT to train the context reasoner in ContextAgent. Specifically, we construct the SFT dataset $\mathcal{D}_{SFT} = \{(\mathcal{X}, \mathcal{T}, \mathcal{Y})\}$. Here, $\mathcal{X}$ contains the sensory context $\mathcal{C}$ and persona context $\mathcal{P}$. The thought traces $\mathcal{T}$ divided by <think> and </think>, are distilled from advanced LLMs [2], enabling ContextAgent to "think before acting", generating explicit thought traces before proactive predictions and tool calls. The output $\mathcal{Y}$ contains proactive scores $\mathcal{P}_{\mathcal{S}}$ and planned tool chains $\mathcal{T}_{\mathcal{C}}$.

## 5 Experiments

### 5.1 Experimental Setup

**Implementation Details.** Our experiments are conducted using 8 A6000 GPUs. For SFT, we use the AdamW optimizer with a learning rate of 0.0001 and apply LoRA techniques during model training. We set the LoRA rank to 8 and use a cosine scheduler with a 10% warmup ratio, training for 5 epochs.

For ICL-based baselines, we randomly select 10 samples from the dataset as demonstrations included in the prompt. We randomly split the dataset into 60% training and 40% testing in our experiments.

**Metrics.** We employ two categories of metrics to evaluate the performance of context-aware proactive LLM agents, including proactive prediction and tool calling. Details of each metric are as follows.

- **Proactive Prediction.** We first evaluate the agent's ability to accurately determine the need for initiating proactive services. Specifically, we use four metrics to assess proactive prediction performance, including the accuracy of proactive predictions (**Acc-P**), missed detections (**MD**), false detections (**FD**), and the root mean square error (**RMSE**) between predicted proactive scores and ground-truth. Acc-P, MD, and FD are commonly used in the existing work [24], while RMSE provides a finer-grained evaluation of the performance of predicted proactive scores.

- **Tool Calling.** To evaluate the agent's tool calling performance, we follow existing works [5, 8] and use standard metrics such as **Precision**, **Recall**, and **F1-score** to compare the tool names in the predicted tool set with those in the ground-truth tool set. We also use **Acc-Args** to evaluate whether the proactive agent can correctly generate the structured data for tool calls, including the tool names and arguments. If an argument of any tool is incorrect, the entire sample is considered incorrect. For Acc-Args, we calculate the accuracy only for the correctly predicted tools to ensure a fair comparison of different approaches.

**Baselines.** We compare ContextAgent with several baselines, including Proactive Agent [24], vanilla ICL, CoT, ICL-P, ICL-All, vanilla SFT, and SFT-P. For the Proactive Agent, we follow [24] and modify the task instructions in the system prompt to adapt to the proactive agent task. For the vanilla ICL, we use few-shot demonstrations with only sensory contexts. For CoT, we include both sensory contexts and thought traces. For ICL-P, we include sensory contexts and personas, and for ICL-All, we incorporate sensory contexts, thought traces, and personas into the prompt. Vanilla SFT uses sensory contexts for fine-tuning. SFT-P uses both sensory and persona contexts. We conduct experiments on 13 LLMs, comprising **(1) proprietary LLMs** including GPT-4o [17] GPT-3.5 [48], GPT-o3, GPT-o4-mini, and Claude Sonnet 4, **(2) open-source LLMs** including Llama-3.1-70B-Instruct [15] and Qwen2.5-72B-Instruct [40], and **(3) small LLMs** including Llama-3.1-8B-Instruct, Qwen2.5-7B-Instruct, Qwen2.5-3B-Instruct, Qwen2.5-1.5B-Instruct, DeepSeek-R1-Distill-Qwen-7B [10], and DeepSeek-R1-Distill-Qwen-1.5B. Details of the baselines and implementation are in the Appendix B.

## 5.2 Results on Benchmarks

**Quantitative Results on ContextAgentBench.** Tab. 1 shows the overall performance of ContextAgent on ContextAgentBench. Results show that when using Llama3.1-8B-Instruct as the base LLM, ContextAgent consistently achieves the highest performance across all metrics, with increases of 8.5% in Acc-P, 7.0% in F1-score, and 6.0% in Acc-Args. Fig. 5 shows that ContextAgent can achieve performance comparable to or even exceeding baselines that employ 70B-scale LLMs and proprietary LLMs, with metrics such as Acc-P (-1.5%), F1-score (-3.0%), and Acc-Args (+6.6%). Due to space constraints, Fig. 5 shows only three key metrics. See Tab. 6 in Appendix E for full comparison.

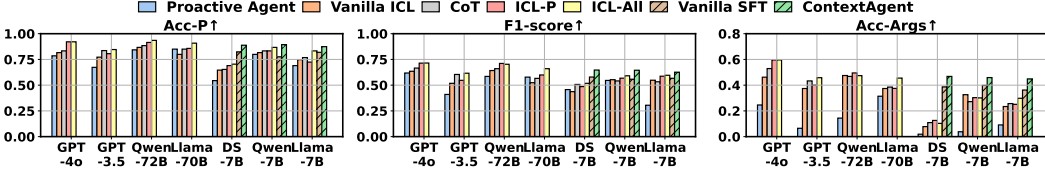

Figure 5: Main results on ContextAgentBench. 'DS' refers to 'DeepSeek'.

**Quantitative Results on ContextAgentBench-Lite.** Fig. 6 and Tab. 2 show the performance of ContextAgent on ContextAgentBench-Lite. The results indicate that both ContextAgent and the baselines exhibit slight performance degradation. However, ContextAgent still achieves the highest performance across all metrics compared to the baselines. When using Qwen2.5-7B-Instruct as the base LLM, ContextAgent achieves improvements of 6.2% Acc-P, 3.0% F1-score, and 7.6% Acc-Args, over the best baseline. It can even achieve comparable and even higher performance than baselines using 70B-scale and proprietary LLMs. Complete results are provided in Tab. 7 within Appendix E.

Table 1: Main results on ContextAgentBench.

| Model | Method | Proactive Predictions | | | | Tool Calling | | | |
|---|---|---|---|---|---|---|---|---|---|
| | | Acc-P↑ | MD.↓ | FD.↓ | RMSE↓ | Precision↑ | Recall↑ | F1-score↑ | Acc-Args↑ |
| Llama-3.1-8B-Ins | Proactive Agent | 0.676 | 0.017 | 0.306 | 1.915 | 0.397 | 0.290 | 0.318 | 0.081 |
| | Vanilla ICL | 0.742 | 0.224 | 0.033 | 1.853 | 0.608 | 0.533 | 0.552 | 0.269 |
| | CoT | 0.699 | 0.278 | 0.023 | 1.960 | 0.590 | 0.539 | 0.551 | 0.209 |
| | ICL-P | 0.742 | 0.242 | 0.015 | 1.922 | 0.608 | 0.553 | 0.567 | 0.262 |
| | ICL-All | 0.757 | 0.229 | 0.012 | 1.872 | 0.631 | 0.565 | 0.582 | 0.270 |
| | Vanilla SFT | 0.813 | 0.068 | 0.117 | 1.572 | 0.609 | 0.581 | 0.580 | 0.405 |
| | *ContextAgent* | 0.874 | 0.030 | 0.095 | 1.408 | 0.660 | 0.627 | 0.626 | 0.448 |
| DeepSeek-R1-7B | Proactive Agent | 0.544 | 0.411 | 0.044 | 3.093 | 0.467 | 0.454 | 0.457 | 0.019 |
| | Vanilla ICL | 0.646 | 0.248 | 0.105 | 2.568 | 0.457 | 0.433 | 0.437 | 0.078 |
| | CoT | 0.653 | 0.319 | 0.027 | 2.760 | 0.528 | 0.501 | 0.507 | 0.109 |
| | ICL-P | 0.690 | 0.227 | 0.081 | 2.466 | 0.518 | 0.479 | 0.486 | 0.126 |
| | ICL-All | 0.704 | 0.268 | 0.0272 | 2.540 | 0.545 | 0.510 | 0.518 | 0.103 |
| | Vanilla SFT | 0.823 | 0.068 | 0.108 | 1.630 | 0.621 | 0.570 | 0.579 | 0.386 |
| | *ContextAgent* | 0.888 | 0.027 | 0.085 | 1.319 | 0.676 | 0.648 | 0.647 | 0.468 |
| Qwen2.5-7B-Ins | Proactive Agent | 0.799 | 0.136 | 0.064 | 2.038 | 0.578 | 0.536 | 0.546 | 0.038 |
| | Vanilla ICL | 0.816 | 0.088 | 0.095 | 1.752 | 0.590 | 0.545 | 0.553 | 0.326 |
| | CoT | 0.833 | 0.085 | 0.081 | 1.790 | 0.585 | 0.527 | 0.541 | 0.272 |
| | ICL-P | 0.833 | 0.091 | 0.074 | 1.819 | 0.610 | 0.556 | 0.568 | 0.303 |
| | ICL-All | 0.867 | 0.088 | 0.044 | 1.721 | 0.635 | 0.577 | 0.591 | 0.301 |
| | Vanilla SFT | 0.775 | 0.088 | 0.136 | 1.774 | 0.589 | 0.551 | 0.558 | 0.398 |
| | *ContextAgent* | 0.894 | 0.013 | 0.091 | 1.264 | 0.672 | 0.644 | 0.645 | 0.459 |

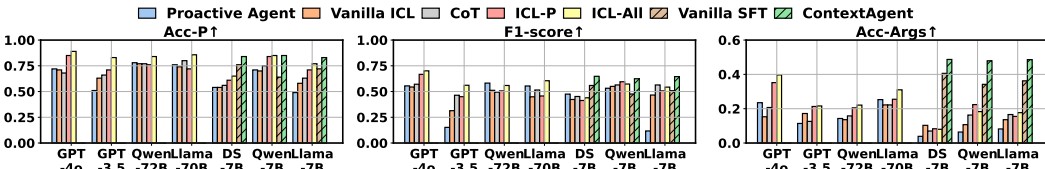

Figure 6: Main results on ContextAgentBench-Lite.

**Qualitative Results.** Fig. 7 and Fig. 8 show the qualitative results of ContextAgent, including both proactive and non-proactive cases. **First**, ContextAgent integrates both sensory and persona contexts for reasoning and offers appropriate proactive services. **Second**, it can further map these contexts to planned tool chains such as the weather and agenda checker, and integrate external knowledge for enhanced proactive service. Fig. 7 shows that ContextAgent can employ GPS, datetime tools, bus schedule checkers, and ride-booking apps like Uber when the user is approaching a bus station. Furthermore, during casual conversations involving proposed outdoor activities, ContextAgent uses tools such as a weather checker, a datetime tool, and an agenda checker to proactively assist the user in evaluating feasibility and making informed decisions. **Third**, ContextAgent can leverage persona context to generate more personalized proactive predictions. For instance, Fig. 7 shows that for health-conscious individuals deciding what to order at a restaurant, ContextAgent proactively offers food-related health information and suggestions. Fig. 8 shows that the persona contexts also help ContextAgent to determine not to disrupt the users. See Appendix E for more qualitative results.

### 5.3 Ablation Study and Discussion

**Impact of Modalities.** We evaluate ContextAgent on ContextAgentBench-Lite to assess sensitivity to missing modalities. Tab. 3 shows that when vision or audio is missing, Acc-P decreases by up to 17.9% and F1-score decreases by up to 23.3%. The results show that both modalities are critical for the context-aware proactive agent, with missing vision having a larger impact than missing audio.
**Sensory Context Perception.** We conduct experiments using the zero-shot Qwen-2.5-VL as the VLM for sensory context extraction in ContextAgent. Tab. 10 shows that this causes ContextAgent to decrease in Acc-P, F1-score, and Acc-Args by 3.0%, 3.3%, and 1.9%, respectively. We observe that the context generated by zero-shot VLM lacks key proactive-oriented cues, such as simply describing the user tying their shoe while sitting on the floor. In contrast, our sensory context extraction module captures detailed scenario information about the gym and specific fitness equipment, providing deeper insight into the user's conditions and intents and resulting in higher performance.
**Persona Context.** We also conduct experiments to study the impact of user personas by removing them during both the training and testing phases. Tab. 10 shows that removing personas consistently

Table 2: Main results on ContextAgentBench-Lite.

| Model | Method | Proactive Predictions | | | | Tool Calling | | | |
|---|---|---|---|---|---|---|---|---|---|
| | | Acc-P$\uparrow$ | MD.$\downarrow$ | FD.$\downarrow$ | RMSE$\downarrow$ | Precision$\uparrow$ | Recall$\uparrow$ | F1-score$\uparrow$ | Acc-Args$\uparrow$ |
| Llama3.1-8B-Ins | Proactive Agent | 0.490 | 0.000 | 0.510 | 2.469 | 0.200 | 0.088 | 0.117 | 0.082 |
| | Vanilla ICL | 0.580 | 0.320 | 0.100 | 2.623 | 0.500 | 0.450 | 0.466 | 0.136 |
| | CoT | 0.630 | 0.360 | 0.010 | 2.306 | 0.595 | 0.553 | 0.564 | 0.166 |
| | ICL-P | 0.710 | 0.210 | 0.080 | 2.315 | 0.535 | 0.495 | 0.506 | 0.155 |
| | ICL-All | 0.770 | 0.170 | 0.060 | 1.757 | 0.598 | 0.526 | 0.543 | 0.177 |
| | Vanilla SFT | 0.720 | 0.120 | 0.160 | 1.959 | 0.536 | 0.497 | 0.508 | 0.362 |
| | SFT-P | 0.734 | 0.115 | 0.151 | 1.980 | 0.555 | 0.582 | 0.552 | 0.353 |
| | *ContextAgent* | 0.830 | 0.070 | 0.100 | 1.510 | 0.687 | 0.637 | 0.645 | 0.486 |
| Qwen2.5-7B-Ins | Proactive Agent | 0.710 | 0.210 | 0.080 | 2.328 | 0.575 | 0.515 | 0.532 | 0.064 |
| | Vanilla ICL | 0.700 | 0.280 | 0.020 | 2.596 | 0.595 | 0.533 | 0.550 | 0.107 |
| | CoT | 0.750 | 0.230 | 0.020 | 2.306 | 0.630 | 0.541 | 0.564 | 0.163 |
| | ICL-P | 0.840 | 0.080 | 0.080 | 1.783 | 0.656 | 0.570 | 0.595 | 0.224 |
| | ICL-All | 0.850 | 0.100 | 0.050 | 1.780 | 0.615 | 0.565 | 0.573 | 0.183 |
| | Vanilla SFT | 0.640 | 0.190 | 0.170 | 2.206 | 0.520 | 0.457 | 0.476 | 0.342 |
| | SFT-P | 0.774 | 0.083 | 0.143 | 1.790 | 0.481 | 0.495 | 0.473 | 0.374 |
| | *ContextAgent* | 0.850 | 0.050 | 0.100 | 1.403 | 0.667 | 0.615 | 0.624 | 0.479 |

Figure 7: Qualitative results of ContextAgent in proactive cases, showing sensory and persona contexts, and ContextAgent's thought traces, proactive predictions, tool calls, and final responses.

leads to significant performance drops, with Llama-3.1-8B-Ins experiencing decreases of up to 9.0% in Acc-P, 12.3% in F1-score, and 12.6% in Acc-Args. Results show that personas are crucial for the proactive agent task, impacting both proactive predictions and tool-calling capabilities.

**Thought Traces.** We also investigate the impact of thought traces on the context-aware proactive agent. We observe that integrating those thought traces can significantly improve ICL performance. Tab. 1 shows that ICL-All achieves up to 20.1% improvement in Acc-P compared to ICL-P, which does not utilize thought traces. Results validate the effectiveness of thought traces for this task. Additionally, as shown in Tab. 10, we also observe that their benefits are reduced under SFT.

**Different Base LLMs and Tool Chain Lengths.** We conduct experiments using different base LLMs in ContextAgent. Tab. 8 shows that Llama-3.1-8B-Instruct and Qwen2.5-7B-Instruct achieve comparable performance and outperform DeepSeek-R1-7B. We also test LLMs in 1.5B to 3B sizes. More details are in Appendix E. In addition, Tab. 11–Tab. 13 show the performance of ContextAgent across samples with varying tool chain lengths. We observe that most approaches achieve higher MD but lower FD, as the prompt we used encourages more conservative initiation of proactive services, leading to less intrusive assistance. See Appendix E for more details.

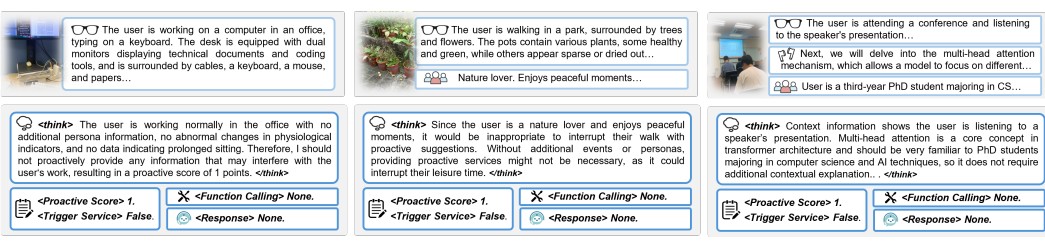

Figure 8: Qualitative results of ContextAgent in non-proactive cases.

Table 3: Performance with missing sensors. "w/o vision" and "w/o audio" denote inference without visual or audio contexts in ContextAgent. The base model is DeepSeek-R1-7B.

| Settings | Proactive Predictions | | | | Tool Calling | | | |
|---|---|---|---|---|---|---|---|---|
| | Acc-P↑ | MD.↓ | FD.↓ | RMSE↓ | Precision↑ | Recall↑ | F1-score↑ | Acc-Args↑ |
| w/o vision | 0.709 | 0.129 | 0.163 | 2.070 | 0.424 | 0.422 | 0.414 | 0.163 |
| w/o audio | 0.720 | 0.173 | 0.106 | 2.020 | 0.501 | 0.501 | 0.493 | 0.212 |
| Full | 0.888 | 0.027 | 0.085 | 1.310 | 0.676 | 0.648 | 0.647 | 0.468 |

## 5.4 Out-of-Domain Evaluation

We also evaluate ContextAgent under an out-of-distribution (OOD) setting. We randomly split ContextAgentBench based on scenarios. Samples from six scenarios are used for training, while those from the remaining three scenarios are used for evaluation. Fig. 9 shows that ContextAgent achieves up to 90.9% Acc-P, 68.9% F1-score, and 51.6% Acc-Args under OOD settings. Furthermore, ContextAgent outperforms the best baseline by 1.9% in Acc-Args, 10.7% in F1-score, and 8.3% in Acc-P, validating its generalization capabilities. Tab. 9 in Appendix E presents the complete results.

Besides, Tab. 4 shows the performance of ContextAgent compared with proprietary and advanced reasoning LLMs under OOD settings. Results demonstrate that ContextAgent achieves comparable performance to these proprietary LLMs. Additionally, enhancing the reasoning capabilities of LLMs can further improve both the accuracy of proactive predictions and tool-calling performance.

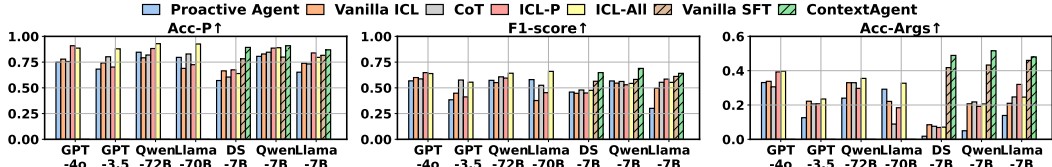

Figure 9: Results on out-of-domain experiments.

Table 4: Performance of ContextAgent compared with proprietary and advanced reasoning LLMs on ContextAgentBench. ContextAgent employs DeepSeek-R1-7B as base model.

| Settings | Proactive Predictions | | | | Tool Calling | | | |
|---|---|---|---|---|---|---|---|---|
| | Acc-P↑ | MD.↓ | FD.↓ | RMSE↓ | Precision↑ | Recall↑ | F1-score↑ | Acc-Args↑ |
| GPT-3.5-Turbo | 0.879 | 0.020 | 0.100 | 1.452 | 0.657 | 0.521 | 0.555 | 0.235 |
| GPT-4o | 0.886 | 0.100 | 0.013 | 1.701 | 0.719 | 0.611 | 0.639 | 0.397 |
| GPT-o4-mini | 0.861 | 0.034 | 0.103 | 1.240 | 0.726 | 0.668 | 0.682 | 0.538 |
| GPT-o3 | 0.868 | 0.069 | 0.062 | 1.100 | 0.755 | 0.697 | 0.711 | 0.563 |
| Claude Sonnet 4 | 0.913 | 0.069 | 0.017 | 1.010 | 0.775 | 0.799 | 0.773 | 0.480 |
| *ContextAgent* | 0.893 | 0.026 | 0.080 | 1.249 | 0.681 | 0.645 | 0.648 | 0.489 |

## 6 Conclusion

This paper introduces ContextAgent, the first framework for context-aware proactive LLM agents. ContextAgent can harness the context information from extensive sensory perceptions and tool-augmented LLM reasoning for enhanced proactive services. To evaluate this new task, we further introduce ContextAgentBench, the first benchmark for evaluating context-aware proactive LLM agents. Our research takes a step towards further aligning with the vision of proactive AI assistants by leveraging rich context from hands-free wearable sensors to enhance proactive LLM reasoning.

## Acknowledgments

The research reported in this paper was partially supported by Research Grants Council of Hong Kong under grants 14207123, STG1/E-403/24-N, and National Science Foundation under Grant Number CNS-1943396. The views and conclusions contained here are those of the authors and should not be interpreted as necessarily representing the official policies or endorsements, either expressed or implied, of The Chinese University of Hong Kong, Columbia University, NSF, or the U.S. Government or any of its agencies.

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

# Appendix

## A    Data Examples

We first provide several examples in our dataset, including cases with different proactive scores.

---

*Example 1*

```
{
    "Context information": "Visual information suggests that
        the user is conversing with a partner on the road.
        Audio information shows there is a conversation between
         the user and others. The partner says "Hi, how about
        organizing a team-building hike this weekend? We haven'
        t had a team-building event in a long time."",

    "Personas": [
        "A software engineer specializing in document
            management systems, working closely with the
            graphic designer to implement the designed
            interfaces."
    ],

    "Thoughts": "When planning hiking activities, various
        factors should be considered, including the weather,
        whether it is the off-season or peak season for tourism
        , passenger flow, schedule, and more. Since the hiking
        location has not been specified, it may be in the city
        where the user is currently located. I need to use GPS
        to determine the user's location.",

    "Proactive score": 5,

    "Tools": "[{"name": "get_current_gps_coordinates", "desc":
        "Get the current GPS coordinates of the user", "params
        ": "None"}, {"name": "get_city_weather", "desc": "Get
        the weather for a specified city at a given time.", "
        params": {"city": "$RESULT(get_current_gps_coordinates.
        city)", "time": "this weekend"}}, {"name": "
        get_current_datetime", "desc": "Get the current date
        and time", "params": "None"}, {"name": "
        check_agenda_time_conflict", "desc": "Check if there is
         a time conflict in the user's agenda for a given
        datetime.", "params": "None"}]",

    "Response": "The weather for the hiking location this
        weekend looks perfect-clear skies and mild temperatures
        . Do you want me to help you pack or suggest what to
        bring?"
}
```

---

*Example 2*

```
{
    "Context information": "Visual information shows the user
        is sitting at a dining table with a plate of food that
        includes pasta, grilled chicken, and a side of salad.",
```

```
    "Personas": [
        "The user has a history of managing Type 2 diabetes.",
        "The user is health-conscious and actively manages
            their diet to control blood sugar levels."
    ],

    "Thoughts": "The user's personas show his glucose level is
        slightly elevated, and the user might appreciate
        dietary advice to help stabilize it. Providing this
        information could be helpful without being intrusive.",

    "Proactive score": 4,

    "Tools": "[{"name": "get_health_data", "desc": "Get health
        data from the user's smart device.", "params": "None
        "},{"name": "wikipedia_search", "desc": "Tool that
        searches the Wikipedia API. Useful for when you need to
         answer general questions about people, places,
        companies, facts, historical events, or other subjects
        .", "params": {"query": "The effect of pasta, grilled
        chicken, and a side of salad on stabilizing blood sugar
         levels."}}]",

    "Response": "Your glucose level appears slightly elevated.
        To help stabilize it, you might consider increasing the
         portion of salad or reducing the amount of pasta.
        Grilled chicken is a great choice for protein, as it
        has minimal impact on blood sugar levels. Would you
        like me to provide additional dietary tips or help
        track your meal for better management?"
}
```

---

*Example 3*

```
{
    "Context information": "Visual information suggests that
        the user is looking in the mirror while wearing summer
        clothes, such as short sleeves and shorts.",

    "Personas": [
        "A university psychology student who is currently
            studying creativity and personality."
    ],

    "Thoughts": "The user is looking in the mirror and may be
        preparing to go out, wearing minimal clothing such as
        short sleeves and shorts. I need to check the weather
        for the user's location. If the temperature is expected
         to drop or if rain is forecasted for the day, I can
        proactively remind the user to dress warmly to avoid
        catching a cold.",

    "Proactive score": 3,

    "Tools": "[{"name": "get_current_gps_coordinates", "desc":
        "Get the current GPS coordinates of the user", "params
        ": "None"},{"name": "get_city_weather", "desc": "Get
        the weather for a specified city at a given time.", "
```

```
        params": {"city": "$RESULT(get_current_gps_coordinates.
        city)", "time":"now"}}]",

    "Response": "I checked the weather forecast, and the local
        temperature is expected to drop sharply after 6 PM.
        Considering that you are only wearing short-sleeved
        shirts and shorts, it is recommended that you add more
        layers."
}
```

---

*Example 4*

```
{
    "Context information": "Visual information suggests the
        user is in a gym, lifting weights.",

    "Personas": [
        "A computer science major interested in developing
            software for audio manipulation and enhancement."
    ],

    "Thoughts": "The user might need guidance on proper
        weightlifting techniques or reminders to take breaks
        between sets. However, without detailed persona
        information, such as their fitness level, I would rate
        this proactive behavior at 2 points. This rating
        indicates some potential for proactivity, but the
        necessity is not very high. After collecting more
        detailed persona information, I could provide more
        accurate proactive services.",

    "Proactive score": 2,
    "Tools": "None",
    "Response": "None"
}
```

---

*Example 5*

```
{
    "Context information": "Visual information shows the user
        is walking on the road. There are many trees on both
        sides of the road, and the road seems to be going
        uphill.",

    "Personas": [
        "A dog owner who wants the best medical treatment for
            their furry friend."
    ],

    "Thoughts": "Contextual information shows user is walking
        on the road. There are many trees on both sides of the
        road, and the road seems to be going uphill. Without
        explicit input or additional contextual information, no
         proactive actions are required. The proactive score is
         set to 1.",

    "Proactive score": 1,
    "Tools": "None",
```

```
        "Response": "None"
}
```

# B  Prompts

This section introduces the details of the prompts used in this work, including the system prompt used in ContextAgent and baselines, the prompt used in the data generation pipeline, and the prompt used in proactive-oriented context extraction.

**System Prompt in ContextAgent.** Fig. 10 shows the prompt used in ContextAgent. It contains both the static prompt and the runtime prompt. The static prompt includes task instructions and toolset definitions that remain constant throughout. The task instructions guide the LLM in understanding its role as a context-aware proactive agent and highlight the key considerations for this task. The toolset definitions allow LLM agents to identify the available external tools and understand how to use them, including their names, arguments, and formats (See Appendix D). The runtime prompt includes user personas and contextual information, which vary across different samples in the dataset.

---

**Prompt Template for ContextAgent**

**## Task Instructions**
You are a proactive LLM agent. You will receive both sensory and persona contexts. The sensory contexts are captured by the multi-modal sensors on the user's wearable devices, such as smart glasses and earphone. Persona context is the personal information like identity and preference. Based on these contexts, you need to generate a thought and determine whether to initiate a proactive service for the user or not. Only propose assistance when you fully understand the user's actions. When generating the response, consider whether you need to call external tools to complete the task. The following are the tool names and their arguments. Please only select tools from the specified tool sets and make sure to use the correct format for their arguments.

**## Tool set**
- GetCityWeather: {*Description*: …, *Arguments*: …}
- CheckAgendaTimeConflict: {*Description*: …, *Arguments*: …}
- ……
- GoogleSearch: {*Description*: …, *Arguments*: …}

Now you will receive the sensory and persona contexts.
**## Persona Context:** { Personas Context }
**## Sensory Context:** { Sensory Context }

---

Figure 10: System prompt for ContextAgent.

**System Prompt for Baselines.** Fig. 11 shows the prompt template used in baseline approaches. To ensure a fair comparison, we keep the task instructions and toolset definitions in the prompt the same as those used in ContextAgent. For the baseline of Proactive Agent, we do not include any samples in the example part of the prompt template. For Vanilla ICL, CoT, ICL-P, and ICL-All, we randomly select ten samples from the training set and incorporate them into the prompt. In Vanilla ICL, the few-shot examples contain only sensory context. In CoT, we additionally include thought traces in the few-shot examples. For ICL-P, we incorporate persona context into the few-shot demonstrations, and for ICL-All, we integrate sensory context, persona context, and thought traces into the few-shot demonstrations. Additionally, for the Vanilla SFT baseline, we use the prompt template shown in Fig. 10, but without including the persona context.

**Prompt for Data Generation Pipeline.** Fig. 12 shows the prompt template used for data generation. We include the required format of data, examples of persona[1] and completed data samples, and detailed descriptions of the accessible tools. Besides, it specifies the structure and formulation of each component within a data sample to guide LLMs to generate high-quality samples. During data generation, we use two strategies, including scenario-aware and proactive score-aware. LLMs are guided to generate samples for different scenarios and target proactive scores separately, which enables LLMs to more effectively learn scenario-specific and score-specific patterns from the provided examples. And when generating samples requiring proactive services, LLMs are instructed to consider

---

[1] https://github.com/tencent-ailab/persona-hub

## *Examples*: {Examples}
Strictly follow the following format for your outputs:
- For proactive samples, the output format should be…
- For not require proactive samples, the output format should be…

Figure 11: Prompt template for baselines.

using tools from our tool set. To further enhance the diversity of the generated samples, we employ various LLMs, including GPT-4o[2], Claude-3.7-Sonnet[3], and Gemini-2.0-Flash[4], for generation.

**Prompt for Proactive-oriented Context Extraction.** Fig. 13 shows the prompt template used for proactive-oriented context extraction. With this prompt, the VLM focuses on objective, detailed scene understanding and key cues capturing (e.g., location, objects, and user actions) that are critical for determining whether proactive assistance is needed. To ensure consistency and output quality, we include five example descriptions within the prompt. These examples illustrate the expected level of detail, structure, and tone, enabling the

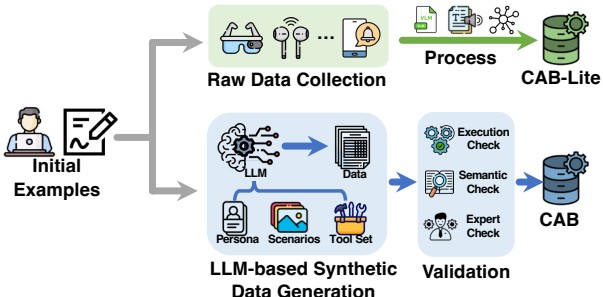

Figure 14: Flowchart illustrating the pipeline for constructing our benchmark. 'CAB' refers to 'ContextAgentBech'.

model to align its output with the expected format. Based on the clear content and formatting guidelines, generated descriptions can provide high-quality context information for the following tasks, including predicting the necessity of proactive services and tool planning. In this study, we employ Qwen-2.5-VL [7] and Whisper [30] to extract the visual and audio contexts, respectively.

## C Dataset Details

As described in Sec. 3.2, the annotators first brainstorm and design initial exemplars to construct the seed dataset. We then utilize an automated diversification pipeline to scale the dataset. Fig. 14 shows the flowchart of our pipeline to generate the ContextAgentBench (CAB) and ContextAgentBench-Lite (CAB-Lite). Additionally, we also collect raw video and audio data based on the textual descriptions of sensory contexts in the dataset. This raw data is sourced from both copyright-free

---

[2] https://openai.com/index/gpt-4o-system-card/
[3] https://www.anthropic.com/claude/sonnet
[4] https://deepmind.google/technologies/gemini/flash/

---

**Prompt for Data Generation**

## *Task Instructions*
You are a professional data generation specialist responsible for creating high-quality training examples for the development of a proactive AI assistant system integrated with smart glasses. Your mission is to generate realistic data samples that accurately represent situations in which the AI assistant should proactively trigger functions or tools to assist users based on contextual perceptions. Contextual perceptions include text descriptions of the physical environment or natural conversations with other people. This information can be captured by sensor data, such as RGB data, audio data, and more.

---

## *Task Objective*:
Generate comprehensive, realistic training examples to determine when an AI assistant should proactively offer assistance and when it should wait for explicit user queries. Consider two important factors when initiating proactive service: contextual information from the physical environment and the user's persona.

---

## *Data Format Requirements*:
Each example must follow this exact structure and include all fields:
"example-[number]": {
"Category": [domain area such as Work, Travel, Health, …, and Others.],
                      ……
"Response": [natural-sounding, helpful response the assistant should provide],
}

---

## *Persona*: When creating personas for examples, use the following samples as a guide. Generated personas should follow this style, concise descriptions that capture key user characteristics, habits or preferences that would influence their interaction with proactive assistance: {Personas Examples}

---

## *Tool Set*: When generating data examples, you should limit the tools used to the following toolset, which includes commonly used tools, AI models, or apps on the user's mobile devices. The following dictionary provides the names of the tools and their functions. {Tool Set}

---

## *Guidelines for Professional Data Generation*:
- First-Person Perspective: All "Vision" descriptions must be from the user's point of view as captured by smart glasses (what the user is looking at, not the user themselves).
        ……
- Logical Consistency: Ensure all fields align logically (e.g., a high proactive score should correspond with actions and responses that provide significant value).

---

## *Example Reference*: {Examples}

---

## **When generating examples, remember these key considerations:**
- Would users genuinely appreciate proactive assistance in this scenario?
        ……
- Do not give strong assumption on "Mobile API data", only limited to information that can be obtained from smartphone api/app notification

---

Please generate {Generated Number} diverse, high-quality examples with proactive score of {Proactive Score} for the {Category} category. Format your response as a valid JSON object with each example following the exact structure specified above.

Figure 12: Prompt template for the data generation pipeline.

internet platforms[5] and our own collections. The self-collected samples are primarily used to gather chitchat scenarios. This study has received IRB approval. All participants offered informed consent before any data were collected. Additionally, we have blurred sensitive regions in the video, such as faces, to ensure participants' privacy.

# D Tool Definition

ContextAgentBench contains 20 tools. Given that the main focus of this study is to bridge the sensory context with tool-based LLM agents, we adopt the definitions of tools from existing work[6]. Tab. 5 provides detailed information about the tools, including their names, tool descriptions, and input and output arguments.

---

[5] https://www.pexels.com/license/
[6] https://github.com/AlibabaResearch/DAMO-ConvAI/tree/main/api-bank/

Table 5: The definition of the 20 tools used in ContextAgentBench.

| Index | Name | Description | Input | Output |
|-------|------|-------------|-------|--------|
| 1 | GetCityWeather | Get the weather for a specified city at a given time. | [text] The city to fetch weather for. [text] The time to fetch weather for. | [text] Weather condition for a specified city at a given time. |
| 2 | DateTime | Get the current date and time. | None. | [text] Current date and time. |
| 3 | CheckAgendaTimeConflict | Check if there is a time conflict in the user's agenda for a given datetime and return all events as a summarized string. | [text] The time to check for conflicts. | [text] A summary of all events and whether there is a conflict. |
| 4 | WikipediaSearch | Search on Wikipedia. | [text] Search query. | [text] Wikipedia search result. |
| 5 | GetCurrentGPS. | Get the current GPS coordinates of the user. | None. | [text] GPS coordinates of the user. |
| 6 | GetOnlineProductPrice | Get the price of a product from an online store. | [text] The name of the product to search for. | [text] The price of the product as a string. |
| 7 | SearchRednote | A platform where people share tips on travel, fitness, cooking, and more, allowing users to search for relevant strategies. | [text] The search query. | [text] The search results from rednote. |
| 8 | VisualLanguageModel | Visual Language Model that can answer the user's questions based on the given image. | [image] Any image. [text] The prompt containing the user's question. | [text] The response from the VLLM. |
| 9 | GoogleMap | Get the route and distance from the current location to the destination using Google Maps API. | [text] The starting location. [text] The destination location. | [text] The route and distance information. |
| 10 | BookUber | Book an Uber ride from the current location to the destination. | [text] The starting location. [text] The destination location. | [text] The Uber ride booking confirmation. |
| 11 | GetHealthData | Get health data from the user's smart device. | None. | [text] The health data as a string. |
| 12 | GetMedicalKnowledge | Get medical expert knowledge from the up-to-date medical knowledge database. | [text] The query string containing the medical topic or symptoms. | [text] The medical expert knowledge as a string. |
| 13 | PlayMusic | Play a song from the user's music library. | None. | [text] The song playing confirmation. |
| 14 | AddtoAgenda | Add an event to the user's agenda. | [text] The name of the event to add. [text] The time of the event. | [text] The confirmation message. |
| 15 | CheckBusSchedule | Check the bus schedule for a specific bus stop. | [text] The name of the bus stop. | [text] The bus schedule information. |
| 16 | GoogleSearch | Search on Google. | [text] Search query. | [text] Description of the search result. |
| 17 | SetTimer | Set a timer for a specific duration. | [text] The duration of the timer. | [text] The timer set confirmation. |
| 18 | QueryStock | This API queries the stock price of a given stock code and date. | [text] The stock code of the given stock. [text] The date of the stock price. | [text] The stock price of the given stock. |
| 19 | AddMeeting | This API allows users to make a reservation for a meeting and store the meeting information (e.g., topic, time, location, attendees) in the database. | [text] The topic of the meeting. [text] The start time of the meeting. [text] The location where the meeting to be held. | [text] Success or failed. |
| 20 | SendEmail | This API for sending email, given the receiver, subject and content. | [text] The receiver address of the email. [text] The subject address of the email. [text] The content of the email. | [text] The status of the email. |

## Task Instructions
You are an advanced first-person vision description specialist. Your task is to analyze video frames and generate detailed, objective descriptions specifically for the "Vision" field in proactive AI assistant training examples. These descriptions represent what the user's smart glasses or other ego camera would capture from their perspective.

## Description Requirements
Your vision descriptions must:
1. Be factual and detailed ......
2. Maintain first-person perspective ......
3. Balance detail with conciseness ......
4. Use clear, straightforward language ......
5. Follow the pattern of the provided examples ......

## Example Style Guide
Your descriptions should follow the style of these examples:
Example-1: "The user is in a room filled with multiple individuals, all seated around a circular table, appearing to be gathered for a meeting."
......
Example-5: "The user is walking near a busy street and looking at a bus stop named 'University.'"

## Key Elements to Include
When describing the frame, focus on:
1. Location and setting - Where the user is (office, store, outdoors, restaurant, etc.)
2. Key objects in view - What items are visible and relevant to understanding the scene
3. People present - If others are visible, their general arrangement and actions
4. Observable user activity - What the user appears to be doing based on what's in view
5. Text or signage - Any readable text that is prominent in the view
6. Spatial arrangement - The general layout of important elements in the scene

## Important Guidelines
- Write in third person, referring to "the user" as shown in the examples
- Keep descriptions to 1-2 concise sentences
- Focus on what is directly observable, not what might be happening
- Include specific named elements when visible (like "bus stop named 'University'")
- It's acceptable to mention what the camera/smart glasses "detects" as in Example-2
- Provide enough context for the proactive system to understand the situation
- Avoid interpretations about the user's intentions or thoughts

Please analyze the provided frame and generate a "Vision" field description following these guidelines and matching the style of the examples.

Figure 13: Prompt template for the proactive-oriented context extraction.

# E   More Results

**Full Results on Benchmark.**   We compare the performance of ContextAgent to baselines using different base LLMs. Since our approach requires model fine-tuning, we implement it only on LLMs with fewer than 7B parameters. However, we also provide the performance of baselines using 70B parameter LLMs and other advanced commercial LLMs as a reference. Tab. 6 shows the full experiment results on ContextAgentBench. Results show that ContextAgent consistently outperforms the baselines when using the same LLMs. We also observe that the baseline approach ICL-All using GPT-4o can achieve the highest performance with 92.1% Acc-P, 71.5% F1-score, and 59.6% Acc-Args. Additionally, ContextAgent with a 7B parameter LLM achieves performance comparable to the best baseline using a 70B LLM, with only 0.4% lower Acc-P, 1.5% lower F1-score, and 0.4% higher Acc-Args, respectively, demonstrating the strong performance of ContextAgent.

Tab. 7 shows the full results on ContextAgentBench-Lite. Tab. 9 shows the full results on ContextAgentBench under OOD settings. Similarly, we observe that ContextAgent consistently achieves the highest performance when using the same base LLMs as the baselines. In addition, ContextAgent can still achieve performance comparable to the baselines that use 70B scale LLMs. For example, ContextAgent using Qwen2.5-7B-Ins can achieve 0.7% lower Acc-P, 1.9% higher F1-score,

Table 6: Main results on ContextAgentBench.

| Model | Method | Proactive Predictions | | | | Tool Calling | | | |
|---|---|---|---|---|---|---|---|---|---|
| | | Acc-P$^\uparrow$ | MD.$^\downarrow$ | FD.$^\downarrow$ | RMSE$^\downarrow$ | Precision$^\uparrow$ | Recall$^\uparrow$ | F1-score$^\uparrow$ | Acc-Args$^\uparrow$ |
| GPT-4o | Proactive Agent | 0.785 | 0.197 | 0.017 | 2.239 | 0.696 | 0.591 | 0.618 | 0.246 |
| | Vanilla ICL | 0.816 | 0.166 | 0.017 | 2.079 | 0.703 | 0.614 | 0.635 | 0.462 |
| | CoT | 0.833 | 0.142 | 0.023 | 2.016 | 0.732 | 0.645 | 0.666 | 0.529 |
| | ICL-P | 0.921 | 0.061 | 0.017 | 1.507 | 0.797 | 0.685 | 0.714 | 0.596 |
| | ICL-All | 0.921 | 0.057 | 0.020 | 1.472 | 0.788 | 0.690 | 0.715 | 0.596 |
| GPT-3.5-Turbo | Proactive Agent | 0.673 | 0.010 | 0.316 | 1.815 | 0.493 | 0.380 | 0.410 | 0.065 |
| | Vanilla ICL | 0.772 | 0.064 | 0.163 | 1.710 | 0.602 | 0.490 | 0.518 | 0.374 |
| | CoT | 0.836 | 0.088 | 0.074 | 1.534 | 0.662 | 0.584 | 0.604 | 0.433 |
| | ICL-P | 0.806 | 0.040 | 0.153 | 1.713 | 0.634 | 0.517 | 0.547 | 0.400 |
| | ICL-All | 0.846 | 0.054 | 0.098 | 1.439 | 0.681 | 0.595 | 0.616 | 0.458 |
| Qwen2.5-72B-Ins | Proactive Agent | 0.843 | 0.064 | 0.091 | 1.717 | 0.670 | 0.555 | 0.585 | 0.144 |
| | Vanilla ICL | 0.867 | 0.078 | 0.054 | 1.324 | 0.695 | 0.625 | 0.642 | 0.475 |
| | CoT | 0.884 | 0.071 | 0.044 | 1.254 | 0.708 | 0.643 | 0.660 | 0.468 |
| | ICL-P | 0.915 | 0.040 | 0.044 | 1.078 | 0.774 | 0.687 | 0.711 | 0.495 |
| | ICL-All | 0.935 | 0.037 | 0.027 | 1.059 | 0.750 | 0.688 | 0.703 | 0.474 |
| Llama3.1-70B-Ins | Proactive Agent | 0.850 | 0.061 | 0.088 | 1.643 | 0.642 | 0.554 | 0.578 | 0.314 |
| | Vanilla ICL | 0.799 | 0.013 | 0.187 | 1.442 | 0.582 | 0.501 | 0.524 | 0.374 |
| | CoT | 0.850 | 0.020 | 0.129 | 1.237 | 0.637 | 0.543 | 0.566 | 0.385 |
| | ICL-P | 0.857 | 0.000 | 0.143 | 1.322 | 0.660 | 0.578 | 0.599 | 0.375 |
| | ICL-All | 0.908 | 0.003 | 0.088 | 1.061 | 0.712 | 0.644 | 0.660 | 0.455 |
| DeepSeek-R1-7B | Proactive Agent | 0.544 | 0.411 | 0.044 | 3.093 | 0.467 | 0.454 | 0.457 | 0.019 |
| | Vanilla ICL | 0.646 | 0.248 | 0.105 | 2.568 | 0.457 | 0.433 | 0.437 | 0.078 |
| | CoT | 0.653 | 0.319 | 0.027 | 2.760 | 0.528 | 0.501 | 0.507 | 0.109 |
| | ICL-P | 0.690 | 0.227 | 0.081 | 2.466 | 0.518 | 0.479 | 0.486 | 0.126 |
| | ICL-All | 0.704 | 0.268 | 0.027 | 2.540 | 0.545 | 0.510 | 0.518 | 0.103 |
| | Vanilla SFT | 0.823 | 0.068 | 0.108 | 1.630 | 0.621 | 0.570 | 0.579 | 0.386 |
| | *ContextAgent* | 0.888 | 0.027 | 0.085 | 1.319 | 0.676 | 0.648 | 0.647 | 0.468 |
| Qwen2.5-7B-Ins | Proactive Agent | 0.799 | 0.136 | 0.064 | 2.038 | 0.578 | 0.536 | 0.546 | 0.038 |
| | Vanilla ICL | 0.816 | 0.088 | 0.095 | 1.752 | 0.590 | 0.545 | 0.553 | 0.326 |
| | CoT | 0.833 | 0.085 | 0.081 | 1.790 | 0.585 | 0.527 | 0.541 | 0.272 |
| | ICL-P | 0.833 | 0.091 | 0.074 | 1.819 | 0.610 | 0.556 | 0.568 | 0.303 |
| | ICL-All | 0.867 | 0.088 | 0.044 | 1.721 | 0.635 | 0.577 | 0.591 | 0.301 |
| | Vanilla SFT | 0.775 | 0.088 | 0.136 | 1.774 | 0.589 | 0.551 | 0.558 | 0.398 |
| | *ContextAgent* | 0.894 | 0.013 | 0.091 | 1.264 | 0.672 | 0.644 | 0.645 | 0.459 |
| Llama3.1-8B-Ins | Proactive Agent | 0.690 | 0.006 | 0.302 | 1.831 | 0.376 | 0.280 | 0.305 | 0.091 |
| | Vanilla ICL | 0.748 | 0.193 | 0.057 | 1.898 | 0.612 | 0.526 | 0.548 | 0.234 |
| | CoT | 0.768 | 0.159 | 0.071 | 1.770 | 0.596 | 0.512 | 0.533 | 0.257 |
| | ICL-P | 0.724 | 0.268 | 0.006 | 2.207 | 0.658 | 0.563 | 0.587 | 0.251 |
| | ICL-All | 0.833 | 0.139 | 0.027 | 1.624 | 0.662 | 0.573 | 0.596 | 0.298 |
| | Vanilla SFT | 0.819 | 0.071 | 0.108 | 1.650 | 0.597 | 0.567 | 0.567 | 0.362 |
| | *ContextAgent* | 0.874 | 0.030 | 0.095 | 1.408 | 0.660 | 0.627 | 0.626 | 0.448 |

and 16.9% higher Acc-Args compared to the best baseline with Llama3.1-70B-Ins. Results validate the effectiveness of ContextAgent.

**Impact of Thought Traces on ICL.** We also analyze the effectiveness of thought traces on ICL. First, we observe that CoT generally outperforms Vanilla ICL on ContextAgentBench, both in in-domain and OOD settings. This indicates that incorporating thought traces into few-shot demonstrations can enhance the performance of ICL. Additionally, we observe that ICL-All also outperforms ICL-P most of the time. This suggests that even after integrating persona context, further incorporating thought traces can offer benefits to ICL approaches for this task. Furthermore, we observe that thought traces provide greater benefits for 70B-sized LLMs compared to 7B LLMs. For instance, using Llama3.1-70B-Ins, ICL-All achieves a 20.1% higher Acc-P, a 20.7% higher F1-score, and a 14.3% higher Acc-Args than ICL-P on ContextAgentBench under OOD settings. This may be because the limited parameters in smaller LLMs result in inherently limited knowledge, making it challenging for them to fully learn the distilled thought traces from more advanced LLMs.

**Impact of Different Base Models.** We test using different LLMs as the base model in ContextAgent. Table 8 shows that 1.5B to 3B LLMs (e.g., Qwen2.5-1.5B-Instruct) perform only 2.9%, 3.9%, and 5.5% lower on Acc-P, F1-score, and Acc-Args, respectively, compared to 7B LLMs. This reveals the opportunities to deploy ContextAgent on mobile devices without accessing the cloud, further reducing privacy concerns and system overhead [41].

Table 7: Main results on ContextAgentBench-Lite.

| Model | Method | Proactive Predictions | | | | Tool Calling | | | |
|---|---|---|---|---|---|---|---|---|---|
| | | Acc-P↑ | MD.↓ | FD.↓ | RMSE↓ | Precision↑ | Recall↑ | F1-score↑ | Acc-Args↑ |
| GPT-4o | Proactive Agent | 0.720 | 0.260 | 0.020 | 2.481 | 0.630 | 0.531 | 0.555 | 0.235 |
| | Vanilla ICL | 0.710 | 0.270 | 0.020 | 2.517 | 0.600 | 0.525 | 0.544 | 0.153 |
| | CoT | 0.680 | 0.310 | 0.010 | 2.630 | 0.607 | 0.557 | 0.571 | 0.207 |
| | ICL-P | 0.850 | 0.130 | 0.020 | 1.770 | 0.755 | 0.635 | 0.667 | 0.352 |
| | ICL-All | 0.890 | 0.110 | 0.000 | 1.627 | 0.782 | 0.675 | 0.701 | 0.397 |
| GPT-3.5-Turbo | Proactive Agent | 0.510 | 0.010 | 0.480 | 2.174 | 0.220 | 0.128 | 0.152 | 0.114 |
| | Vanilla ICL | 0.630 | 0.050 | 0.320 | 2.253 | 0.410 | 0.282 | 0.314 | 0.172 |
| | CoT | 0.660 | 0.190 | 0.150 | 2.186 | 0.531 | 0.439 | 0.464 | 0.126 |
| | ICL-P | 0.710 | 0.020 | 0.270 | 1.854 | 0.545 | 0.415 | 0.450 | 0.213 |
| | ICL-All | 0.830 | 0.060 | 0.110 | 1.578 | 0.635 | 0.537 | 0.561 | 0.216 |
| Qwen2.5-72B-Ins | Proactive Agent | 0.780 | 0.180 | 0.040 | 2.258 | 0.670 | 0.550 | 0.582 | 0.143 |
| | Vanilla ICL | 0.770 | 0.120 | 0.110 | 2.000 | 0.595 | 0.481 | 0.512 | 0.136 |
| | CoT | 0.770 | 0.120 | 0.110 | 1.786 | 0.575 | 0.467 | 0.494 | 0.158 |
| | ICL-P | 0.760 | 0.030 | 0.210 | 1.612 | 0.605 | 0.471 | 0.506 | 0.205 |
| | ICL-All | 0.840 | 0.020 | 0.140 | 1.349 | 0.646 | 0.530 | 0.559 | 0.221 |
| Llama3.1-70B-Ins | Proactive Agent | 0.760 | 0.150 | 0.090 | 2.076 | 0.620 | 0.529 | 0.554 | 0.253 |
| | Vanilla ICL | 0.740 | 0.050 | 0.210 | 1.889 | 0.530 | 0.421 | 0.449 | 0.222 |
| | CoT | 0.800 | 0.060 | 0.140 | 1.649 | 0.585 | 0.491 | 0.515 | 0.222 |
| | ICL-P | 0.720 | 0.010 | 0.270 | 1.841 | 0.546 | 0.426 | 0.457 | 0.255 |
| | ICL-All | 0.857 | 0.035 | 0.107 | 1.300 | 0.684 | 0.581 | 0.605 | 0.310 |
| DeepSeek-R1-7B | Proactive Agent | 0.540 | 0.420 | 0.040 | 3.119 | 0.490 | 0.470 | 0.475 | 0.039 |
| | Vanilla ICL | 0.540 | 0.310 | 0.150 | 2.849 | 0.455 | 0.411 | 0.423 | 0.103 |
| | CoT | 0.560 | 0.320 | 0.120 | 2.796 | 0.465 | 0.447 | 0.452 | 0.070 |
| | ICL-P | 0.610 | 0.240 | 0.150 | 2.624 | 0.445 | 0.400 | 0.413 | 0.083 |
| | ICL-All | 0.650 | 0.260 | 0.090 | 2.541 | 0.455 | 0.434 | 0.439 | 0.080 |
| | Vanilla SFT | 0.760 | 0.120 | 0.120 | 1.786 | 0.581 | 0.561 | 0.559 | 0.406 |
| | *ContextAgent* | 0.840 | 0.050 | 0.110 | 1.510 | 0.678 | 0.641 | 0.648 | 0.487 |
| Qwen2.5-7B-Ins | Proactive Agent | 0.710 | 0.210 | 0.080 | 2.328 | 0.575 | 0.515 | 0.532 | 0.064 |
| | Vanilla ICL | 0.700 | 0.280 | 0.020 | 2.596 | 0.595 | 0.533 | 0.550 | 0.107 |
| | CoT | 0.750 | 0.230 | 0.020 | 2.306 | 0.630 | 0.541 | 0.564 | 0.163 |
| | ICL-P | 0.840 | 0.080 | 0.080 | 1.783 | 0.656 | 0.570 | 0.595 | 0.224 |
| | ICL-All | 0.850 | 0.100 | 0.050 | 1.780 | 0.615 | 0.565 | 0.573 | 0.183 |
| | Vanilla SFT | 0.640 | 0.190 | 0.170 | 2.206 | 0.520 | 0.457 | 0.476 | 0.342 |
| | *ContextAgent* | 0.850 | 0.050 | 0.100 | 1.403 | 0.667 | 0.615 | 0.624 | 0.479 |
| Llama3.1-8B-Ins | Proactive Agent | 0.490 | 0.000 | 0.510 | 2.469 | 0.200 | 0.088 | 0.117 | 0.082 |
| | Vanilla ICL | 0.580 | 0.320 | 0.100 | 2.623 | 0.500 | 0.450 | 0.466 | 0.136 |
| | CoT | 0.630 | 0.360 | 0.010 | 2.306 | 0.595 | 0.553 | 0.564 | 0.166 |
| | ICL-P | 0.710 | 0.210 | 0.080 | 2.315 | 0.535 | 0.495 | 0.506 | 0.155 |
| | ICL-All | 0.770 | 0.170 | 0.060 | 1.757 | 0.598 | 0.526 | 0.543 | 0.177 |
| | Vanilla SFT | 0.720 | 0.120 | 0.160 | 1.959 | 0.536 | 0.497 | 0.508 | 0.362 |
| | *ContextAgent* | 0.830 | 0.070 | 0.100 | 1.510 | 0.687 | 0.637 | 0.645 | 0.486 |

Table 8: Overall performance of ContextAgent using different LLMs as base models.

| Category | Model | Size | Proactive Predictions | | | | Tool Calling | | | |
|---|---|---|---|---|---|---|---|---|---|---|
| | | | Acc-P↑ | MD.↓ | FD.↓ | RMSE↓ | Precision↑ | Recall↑ | F1-score↑ | Acc-Args↑ |
| 7B ∼ 8B | LLaMA3 | 8B | 0.898 | 0.028 | 0.074 | 1.254 | 0.685 | 0.653 | 0.652 | 0.465 |
| | Qwen2.5 | 7B | 0.883 | 0.040 | 0.076 | 1.215 | 0.682 | 0.653 | 0.653 | 0.481 |
| | DeepSeek-R1 | 7B | 0.888 | 0.038 | 0.074 | 1.275 | 0.659 | 0.648 | 0.639 | 0.434 |
| 1.5B ∼ 3B | Qwen2.5 | 3B | 0.869 | 0.043 | 0.086 | 1.336 | 0.652 | 0.610 | 0.615 | 0.421 |
| | DeepSeek-R1 | 1.5B | 0.882 | 0.041 | 0.076 | 1.245 | 0.686 | 0.652 | 0.652 | 0.447 |
| | Qwen2.5 | 1.5B | 0.869 | 0.038 | 0.091 | 1.312 | 0.642 | 0.612 | 0.613 | 0.410 |

Tab. 10 shows the ablation study using different base LLMs. The results show that persona context is crucial for the task. Removing it from ContextAgent can lead to significant decreases across all metrics, with Acc-P and Acc-Args reductions of up to 12.0% and 14.3%, respectively. Additionally, sensory context perception and thought traces can also bring positive benefits.

**Performance Across Different Tool Chain Lengths.** Although Tab. 6 shows that ContextAgent outperforms the baselines on ContextAgentBench, we provide a detailed analysis in this section. Specifically, we examine the performance of ContextAgent across samples with varying tool chain lengths, categorizing them into three groups: 0–1 tools (level 1), 2 tools (level 2), and 3–5 tools (level

Table 9: Results on out-of-domain experiments.

| Model | Method | Proactive Predictions | | | | Tool Calling | | | |
|---|---|---|---|---|---|---|---|---|---|
| | | Acc-P$^\uparrow$ | MD.$^\downarrow$ | FD.$^\downarrow$ | RMSE$^\downarrow$ | Precision$^\uparrow$ | Recall$^\uparrow$ | F1-score$^\uparrow$ | Acc-Args$^\uparrow$ |
| GPT-4o | Proactive Agent | 0.749 | 0.234 | 0.016 | 2.446 | 0.652 | 0.542 | 0.568 | 0.331 |
| | Vanilla ICL | 0.779 | 0.207 | 0.013 | 2.253 | 0.681 | 0.573 | 0.600 | 0.338 |
| | CoT | 0.756 | 0.234 | 0.010 | 2.407 | 0.659 | 0.565 | 0.587 | 0.306 |
| | ICL-P | 0.909 | 0.076 | 0.013 | 1.514 | 0.735 | 0.619 | 0.648 | 0.393 |
| | ICL-All | 0.886 | 0.100 | 0.013 | 1.701 | 0.719 | 0.611 | 0.639 | 0.397 |
| GPT-3.5-Turbo | Proactive Agent | 0.682 | 0.010 | 0.307 | 1.757 | 0.478 | 0.352 | 0.384 | 0.126 |
| | Vanilla ICL | 0.742 | 0.050 | 0.207 | 1.797 | 0.555 | 0.412 | 0.448 | 0.222 |
| | CoT | 0.802 | 0.183 | 0.013 | 2.152 | 0.658 | 0.549 | 0.576 | 0.206 |
| | ICL-P | 0.702 | 0.000 | 0.297 | 1.589 | 0.521 | 0.376 | 0.412 | 0.207 |
| | ICL-All | 0.879 | 0.020 | 0.100 | 1.452 | 0.657 | 0.521 | 0.555 | 0.235 |
| Qwen2.5-72B-Ins | Proactive Agent | 0.846 | 0.070 | 0.083 | 1.806 | 0.657 | 0.545 | 0.573 | 0.240 |
| | Vanilla ICL | 0.792 | 0.090 | 0.117 | 1.600 | 0.627 | 0.522 | 0.551 | 0.330 |
| | CoT | 0.819 | 0.140 | 0.040 | 1.564 | 0.678 | 0.583 | 0.607 | 0.330 |
| | ICL-P | 0.882 | 0.013 | 0.103 | 1.199 | 0.675 | 0.569 | 0.595 | 0.297 |
| | ICL-All | 0.929 | 0.020 | 0.050 | 1.036 | 0.717 | 0.618 | 0.642 | 0.355 |
| Llama3.1-70B-Ins | Proactive Agent | 0.796 | 0.107 | 0.097 | 1.950 | 0.659 | 0.551 | 0.580 | 0.292 |
| | Vanilla ICL | 0.689 | 0.020 | 0.291 | 1.840 | 0.445 | 0.350 | 0.377 | 0.221 |
| | CoT | 0.829 | 0.026 | 0.143 | 1.386 | 0.629 | 0.489 | 0.525 | 0.089 |
| | ICL-P | 0.725 | 0.003 | 0.271 | 1.734 | 0.533 | 0.423 | 0.453 | 0.184 |
| | ICL-All | 0.926 | 0.003 | 0.070 | 1.127 | 0.756 | 0.626 | 0.660 | 0.327 |
| DeepSeek-R1-7B | Proactive Agent | 0.571 | 0.391 | 0.036 | 3.040 | 0.475 | 0.453 | 0.459 | 0.018 |
| | Vanilla ICL | 0.665 | 0.254 | 0.080 | 2.621 | 0.483 | 0.435 | 0.447 | 0.085 |
| | CoT | 0.605 | 0.347 | 0.046 | 2.777 | 0.516 | 0.467 | 0.479 | 0.076 |
| | ICL-P | 0.675 | 0.237 | 0.087 | 2.527 | 0.489 | 0.438 | 0.450 | 0.069 |
| | ICL-All | 0.639 | 0.321 | 0.040 | 2.637 | 0.505 | 0.466 | 0.476 | 0.071 |
| | Vanilla SFT | 0.782 | 0.087 | 0.130 | 1.743 | 0.585 | 0.565 | 0.564 | 0.418 |
| | *ContextAgent* | 0.893 | 0.026 | 0.080 | 1.249 | 0.681 | 0.645 | 0.648 | 0.489 |
| Qwen2.5-7B-Ins | Proactive Agent | 0.806 | 0.147 | 0.046 | 2.139 | 0.605 | 0.556 | 0.567 | 0.050 |
| | Vanilla ICL | 0.829 | 0.120 | 0.050 | 1.898 | 0.597 | 0.532 | 0.546 | 0.207 |
| | CoT | 0.846 | 0.123 | 0.030 | 1.836 | 0.612 | 0.545 | 0.562 | 0.218 |
| | ICL-P | 0.886 | 0.050 | 0.063 | 1.584 | 0.569 | 0.517 | 0.530 | 0.192 |
| | ICL-All | 0.890 | 0.040 | 0.070 | 1.502 | 0.593 | 0.527 | 0.543 | 0.206 |
| | Vanilla SFT | 0.799 | 0.077 | 0.123 | 1.685 | 0.607 | 0.585 | 0.582 | 0.433 |
| | *ContextAgent* | 0.909 | 0.020 | 0.070 | 1.172 | 0.711 | 0.699 | 0.689 | 0.516 |
| Llama3.1-8B-Ins | Proactive Agent | 0.652 | 0.013 | 0.334 | 1.918 | 0.403 | 0.266 | 0.301 | 0.139 |
| | Vanilla ICL | 0.739 | 0.163 | 0.097 | 2.052 | 0.565 | 0.470 | 0.496 | 0.210 |
| | CoT | 0.732 | 0.220 | 0.046 | 1.934 | 0.623 | 0.529 | 0.554 | 0.247 |
| | ICL-P | 0.839 | 0.137 | 0.023 | 1.907 | 0.659 | 0.558 | 0.585 | 0.320 |
| | ICL-All | 0.796 | 0.173 | 0.030 | 1.822 | 0.623 | 0.527 | 0.553 | 0.246 |
| | Vanilla SFT | 0.816 | 0.090 | 0.093 | 1.161 | 0.628 | 0.619 | 0.611 | 0.460 |
| | *ContextAgent* | 0.869 | 0.036 | 0.093 | 1.369 | 0.661 | 0.645 | 0.641 | 0.480 |

3), as shown in Tab. 11, Tab. 12, and Tab. 13, respectively. Results show that ContextAgent achieves significantly higher F1-score, and Acc-Args than the baselines on the level 2 and level 3 samples. For level 3 samples, ContextAgent can achieve 30.3% and 16.0% higher Acc-P and Acc-Args, respectively, when using Qwen2.5-7B-Ins as the base LLM, validating its strong performance in scenarios that require multiple external tools for proactive services.

**More Qualitative Results.** We also provide more qualitative results of ContextAgent in Fig. 15 and Fig. 16. Results indicate that ContextAgent initiates the proactive support when it perceives contexts such as the user showing interest in a product or putting clothes into a washing machine. Additionally, integrating knowledge from external tools, such as online product prices and app-based recommendations, can further enhance the usefulness of these proactive services. Moreover, when ContextAgent detects contexts like leisure time enjoying the sunset or beach waves, or normal daily activities such as walking upstairs, it correctly identifies that no proactive support is needed, thus reducing unnecessary interruptions.

# F Limitations

Although ContextAgent demonstrates strong performance in the context-aware proactive agent task, we summarize the limitations and future directions for this work as follows. First, the current tool set in ContextAgent includes twenty tools with APIs. The recent emergence of the Model Context

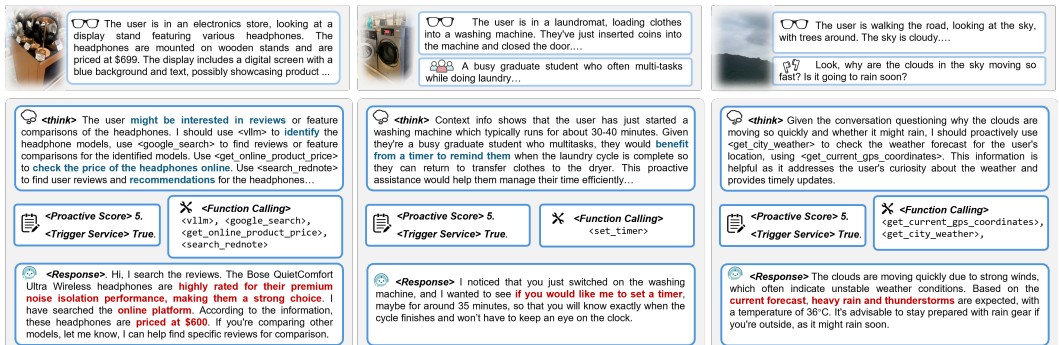

Figure 15: Qualitative results of ContextAgent in proactive cases, showing sensory and persona contexts, and ContextAgent's thought traces, proactive predictions, tool calls, and final responses.

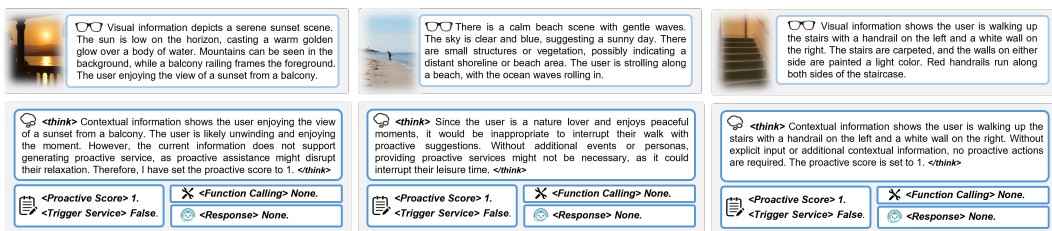

Figure 16: Qualitative results of ContextAgent in non-proactive cases.

Protocol (MCP) aims to standardize these external tools, which can be integrated into ContextAgent to further enhance the standardization of tool calls and the diversity of tool types. Second, although ContextAgentBench currently includes nine life scenarios, it can be further diversified to enhance the ContextAgent's capabilities and practicality for daily use.

# G    Broader Impacts

In this paper, we explore the context-aware proactive LLM agents for the first time, and propose a framework called ContextAgent. By utilizing the rich contextual information from sensory perceptions alongside tool-based LLM reasoning, ContextAgent significantly enhances both perception and functionality compared to existing approaches, resulting in improved proactive service. ContextAgent utilizes sensor data from wearable devices such as smart glasses and earphones. This hands-free, egocentric perception not only offers a better understanding of the user's conditions and intentions but also reduces both cognitive and physical workload, perfectly aligning with the vision of a proactive assistant. In addition, to bridge the gap in evaluating this new task, we introduce ContextAgentBench, the first benchmark to evaluate context-aware proactive LLM agents. Furthermore, ContextAgent serves as a bridge between research on sensory context perception in ubiquitous mobile systems and the emerging LLM agents, thereby opening up new research perspectives and directions. We hope our research will help advance the development of proactive, human-centric AI assistants.

Table 10: Ablation study on ContextAgentBench. "w/o persona" means the agent does not use persona information during both the training and testing stages. "w/o think" means that the SFT training data does not include the thought process.

| Model | Method | Proactive Predictions | | | | Tool Calling | | | |
|---|---|---|---|---|---|---|---|---|---|
| | | Acc-P$^\uparrow$ | MD.$^\downarrow$ | FD.$^\downarrow$ | RMSE$^\downarrow$ | Precision$^\uparrow$ | Recall$^\uparrow$ | F1-score$^\uparrow$ | Acc-Args$^\uparrow$ |
| Qwen2.5-7B | w/o persona | 0.775 | 0.078 | 0.146 | 1.799 | 0.571 | 0.531 | 0.532 | 0.364 |
| | w/o think | 0.857 | 0.030 | 0.112 | 1.415 | 0.634 | 0.592 | 0.599 | 0.411 |
| | ContextAgent | 0.894 | 0.013 | 0.091 | 1.264 | 0.672 | 0.644 | 0.645 | 0.459 |
| Llama3.1-8B | w/o persona | 0.806 | 0.081 | 0.112 | 1.639 | 0.611 | 0.574 | 0.579 | 0.405 |
| | w/o think | 0.857 | 0.030 | 0.112 | 1.397 | 0.645 | 0.607 | 0.612 | 0.419 |
| | ContextAgent | 0.874 | 0.030 | 0.095 | 1.408 | 0.660 | 0.627 | 0.626 | 0.448 |
| DeepSeek-R1-7B | w/o persona | 0.799 | 0.092 | 0.109 | 1.742 | 0.609 | 0.575 | 0.580 | 0.409 |
| | w/o think | 0.884 | 0.017 | 0.099 | 1.355 | 0.652 | 0.625 | 0.625 | 0.439 |
| | ContextAgent | 0.888 | 0.027 | 0.085 | 1.319 | 0.676 | 0.648 | 0.647 | 0.468 |

Table 11: Results for the Level-1 samples in ContextAgentBench.

| Model | Method | Proactive Predictions | | | | Tool Calling | | | |
|---|---|---|---|---|---|---|---|---|---|
| | | Acc-P$^\uparrow$ | MD.$^\downarrow$ | FD.$^\downarrow$ | RMSE$^\downarrow$ | Precision$^\uparrow$ | Recall$^\uparrow$ | F1-score$^\uparrow$ | Acc-Args$^\uparrow$ |
| GPT-4o | Proactive Agent | 0.835 | 0.138 | 0.025 | 1.983 | 0.789 | 0.789 | 0.789 | 0.246 |
| | Vanilla ICL | 0.835 | 0.138 | 0.025 | 1.997 | 0.776 | 0.784 | 0.778 | 0.462 |
| | CoT | 0.835 | 0.128 | 0.035 | 1.970 | 0.775 | 0.784 | 0.777 | 0.529 |
| | ICL-P | 0.928 | 0.046 | 0.025 | 1.432 | 0.860 | 0.866 | 0.862 | 0.596 |
| | ICL-All | 0.923 | 0.046 | 0.030 | 1.421 | 0.839 | 0.851 | 0.842 | 0.596 |
| GPT-3.5-Turbo | Proactive Agent | 0.517 | 0.005 | 0.476 | 2.104 | 0.467 | 0.467 | 0.467 | 0.065 |
| | Vanilla ICL | 0.702 | 0.051 | 0.246 | 1.820 | 0.617 | 0.620 | 0.617 | 0.374 |
| | CoT | 0.815 | 0.071 | 0.112 | 1.525 | 0.706 | 0.717 | 0.710 | 0.433 |
| | ICL-P | 0.733 | 0.035 | 0.230 | 1.872 | 0.625 | 0.630 | 0.627 | 0.400 |
| | ICL-All | 0.800 | 0.051 | 0.148 | 1.542 | 0.709 | 0.717 | 0.712 | 0.458 |
| Qwen2.5-72B-Ins | Proactive Agent | 0.800 | 0.061 | 0.069 | 1.684 | 0.700 | 0.702 | 0.700 | 0.144 |
| | Vanilla ICL | 0.835 | 0.082 | 0.082 | 1.353 | 0.729 | 0.743 | 0.734 | 0.475 |
| | CoT | 0.856 | 0.076 | 0.066 | 1.254 | 0.747 | 0.759 | 0.751 | 0.468 |
| | ICL-P | 0.887 | 0.046 | 0.066 | 1.081 | 0.805 | 0.810 | 0.806 | 0.495 |
| | ICL-All | 0.923 | 0.035 | 0.041 | 0.981 | 0.806 | 0.815 | 0.808 | 0.474 |
| Llama3.1-70B-Ins | Proactive Agent | 0.825 | 0.041 | 0.133 | 1.676 | 0.687 | 0.692 | 0.688 | 0.314 |
| | Vanilla ICL | 0.697 | 0.020 | 0.282 | 1.688 | 0.548 | 0.553 | 0.550 | 0.374 |
| | CoT | 0.774 | 0.030 | 0.194 | 1.394 | 0.612 | 0.625 | 0.616 | 0.385 |
| | ICL-P | 0.784 | 0.000 | 0.215 | 1.529 | 0.654 | 0.666 | 0.658 | 0.374 |
| | ICL-All | 0.861 | 0.005 | 0.133 | 1.176 | 0.736 | 0.748 | 0.740 | 0.455 |
| DeepSeek-R1-7B | Proactive Agent | 0.707 | 0.225 | 0.066 | 2.444 | 0.671 | 0.671 | 0.671 | 0.019 |
| | Vanilla ICL | 0.697 | 0.143 | 0.159 | 2.215 | 0.588 | 0.594 | 0.590 | 0.078 |
| | CoT | 0.759 | 0.200 | 0.041 | 2.328 | 0.705 | 0.707 | 0.706 | 0.109 |
| | ICL-P | 0.759 | 0.117 | 0.123 | 2.065 | 0.658 | 0.666 | 0.660 | 0.126 |
| | ICL-All | 0.825 | 0.133 | 0.041 | 1.978 | 0.714 | 0.717 | 0.715 | 0.103 |
| | Vanilla SFT | 0.779 | 0.056 | 0.164 | 1.783 | 0.628 | 0.651 | 0.635 | 0.386 |
| | *ContextAgent* | 0.856 | 0.015 | 0.128 | 1.423 | 0.712 | 0.748 | 0.723 | 0.468 |
| Qwen2.5-7B-Ins | Proactive Agent | 0.820 | 0.082 | 0.097 | 1.843 | 0.692 | 0.702 | 0.695 | 0.038 |
| | Vanilla ICL | 0.769 | 0.087 | 0.143 | 1.868 | 0.642 | 0.661 | 0.648 | 0.326 |
| | CoT | 0.794 | 0.082 | 0.123 | 1.874 | 0.668 | 0.676 | 0.670 | 0.272 |
| | ICL-P | 0.805 | 0.082 | 0.112 | 1.845 | 0.683 | 0.697 | 0.688 | 0.303 |
| | ICL-All | 0.851 | 0.082 | 0.066 | 1.724 | 0.735 | 0.748 | 0.739 | 0.301 |
| | Vanilla SFT | 0.733 | 0.061 | 0.205 | 1.905 | 0.588 | 0.605 | 0.593 | 0.398 |
| | *ContextAgent* | 0.856 | 0.005 | 0.138 | 1.396 | 0.701 | 0.728 | 0.709 | 0.459 |
| Llama3.1-8B-Ins | Proactive Agent | 0.533 | 0.010 | 0.456 | 2.169 | 0.328 | 0.328 | 0.328 | 0.091 |
| | Vanilla ICL | 0.794 | 0.117 | 0.087 | 1.593 | 0.707 | 0.707 | 0.707 | 0.234 |
| | CoT | 0.774 | 0.117 | 0.107 | 1.637 | 0.659 | 0.661 | 0.659 | 0.257 |
| | ICL-P | 0.794 | 0.194 | 0.010 | 1.900 | 0.753 | 0.753 | 0.753 | 0.251 |
| | ICL-All | 0.861 | 0.097 | 0.041 | 1.379 | 0.748 | 0.748 | 0.748 | 0.298 |
| | Vanilla SFT | 0.774 | 0.061 | 0.164 | 1.778 | 0.635 | 0.671 | 0.647 | 0.362 |
| | *ContextAgent* | 0.841 | 0.015 | 0.143 | 1.485 | 0.695 | 0.733 | 0.707 | 0.448 |

Table 12: Results for the Level-2 samples in ContextAgentBench.

| Model | Method | Proactive Predictions | | | | Tool Calling | | | |
|---|---|---|---|---|---|---|---|---|---|
| | | Acc-P$\uparrow$ | MD.$\downarrow$ | FD.$\downarrow$ | RMSE$\downarrow$ | Precision$\uparrow$ | Recall$\uparrow$ | F1-score$\uparrow$ | Acc-Args$\uparrow$ |
| GPT-4o | Proactive Agent | 0.651 | 0.348 | 0.000 | 2.761 | 0.500 | 0.255 | 0.337 | 0.256 |
| | Vanilla ICL | 0.674 | 0.325 | 0.000 | 2.676 | 0.333 | 0.209 | 0.249 | 0.467 |
| | CoT | 0.720 | 0.279 | 0.000 | 2.645 | 0.472 | 0.325 | 0.370 | 0.534 |
| | ICL-P | 0.883 | 0.116 | 0.000 | 1.855 | 0.511 | 0.290 | 0.364 | 0.606 |
| | ICL-All | 0.883 | 0.116 | 0.000 | 1.848 | 0.546 | 0.314 | 0.391 | 0.590 |
| GPT-3.5-Turbo | Proactive Agent | 0.953 | 0.046 | 0.000 | 1.229 | 0.581 | 0.290 | 0.387 | 0.070 |
| | Vanilla ICL | 0.814 | 0.186 | 0.000 | 1.867 | 0.407 | 0.220 | 0.282 | 0.366 |
| | CoT | 0.790 | 0.209 | 0.000 | 1.994 | 0.480 | 0.302 | 0.359 | 0.449 |
| | ICL-P | 0.883 | 0.116 | 0.000 | 1.830 | 0.534 | 0.279 | 0.364 | 0.408 |
| | ICL-All | 0.883 | 0.116 | 0.000 | 1.532 | 0.492 | 0.325 | 0.379 | 0.472 |
| Qwen2.5-72B-Ins | Proactive Agent | 0.930 | 0.069 | 0.000 | 1.758 | 0.534 | 0.302 | 0.379 | 0.150 |
| | Vanilla ICL | 0.883 | 0.116 | 0.000 | 1.509 | 0.484 | 0.372 | 0.405 | 0.484 |
| | CoT | 0.907 | 0.093 | 0.000 | 1.430 | 0.511 | 0.372 | 0.418 | 0.478 |
| | ICL-P | 0.930 | 0.069 | 0.000 | 1.355 | 0.639 | 0.476 | 0.527 | 0.506 |
| | ICL-All | 0.907 | 0.093 | 0.000 | 1.486 | 0.542 | 0.453 | 0.479 | 0.493 |
| Llama3.1-70B-Ins | Proactive Agent | 0.837 | 0.162 | 0.000 | 1.935 | 0.503 | 0.302 | 0.367 | 0.333 |
| | Vanilla ICL | 1.000 | 0.000 | 0.000 | 0.849 | 0.531 | 0.395 | 0.439 | 0.381 |
| | CoT | 1.000 | 0.000 | 0.000 | 0.902 | 0.589 | 0.348 | 0.427 | 0.392 |
| | ICL-P | 1.000 | 0.000 | 0.000 | 0.821 | 0.624 | 0.418 | 0.486 | 0.385 |
| | ICL-All | 1.000 | 0.000 | 0.000 | 0.876 | 0.662 | 0.476 | 0.532 | 0.465 |
| DeepSeek-R1-7B | Proactive Agent | 0.279 | 0.720 | 0.000 | 3.882 | 0.058 | 0.034 | 0.042 | 0.021 |
| | Vanilla ICL | 0.465 | 0.534 | 0.000 | 3.348 | 0.129 | 0.104 | 0.110 | 0.082 |
| | CoT | 0.348 | 0.651 | 0.000 | 3.592 | 0.104 | 0.081 | 0.089 | 0.115 |
| | ICL-P | 0.465 | 0.534 | 0.000 | 3.306 | 0.089 | 0.058 | 0.067 | 0.129 |
| | ICL-All | 0.418 | 0.581 | 0.000 | 3.474 | 0.127 | 0.081 | 0.096 | 0.096 |
| | Vanilla SFT | 0.860 | 0.139 | 0.000 | 1.486 | 0.472 | 0.337 | 0.377 | 0.404 |
| | *ContextAgent* | 0.930 | 0.069 | 0.000 | 1.181 | 0.536 | 0.441 | 0.464 | 0.471 |
| Qwen2.5-7B-Ins | Proactive Agent | 0.674 | 0.325 | 0.000 | 2.663 | 0.302 | 0.220 | 0.248 | 0.040 |
| | Vanilla ICL | 0.907 | 0.093 | 0.000 | 1.372 | 0.461 | 0.383 | 0.408 | 0.324 |
| | CoT | 0.930 | 0.069 | 0.000 | 1.422 | 0.418 | 0.279 | 0.325 | 0.270 |
| | ICL-P | 0.860 | 0.139 | 0.000 | 1.861 | 0.434 | 0.314 | 0.351 | 0.302 |
| | ICL-All | 0.883 | 0.116 | 0.000 | 1.758 | 0.476 | 0.314 | 0.368 | 0.316 |
| | Vanilla SFT | 0.744 | 0.255 | 0.000 | 1.823 | 0.412 | 0.372 | 0.377 | 0.419 |
| | *ContextAgent* | 0.977 | 0.023 | 0.000 | 0.927 | 0.542 | 0.465 | 0.479 | 0.479 |
| Llama3.1-8B-Ins | Proactive Agent | 1.000 | 0.000 | 0.000 | 0.940 | 0.383 | 0.197 | 0.259 | 0.088 |
| | Vanilla ICL | 0.558 | 0.441 | 0.000 | 2.583 | 0.290 | 0.162 | 0.205 | 0.231 |
| | CoT | 0.674 | 0.325 | 0.000 | 2.151 | 0.352 | 0.209 | 0.255 | 0.252 |
| | ICL-P | 0.534 | 0.465 | 0.000 | 2.672 | 0.418 | 0.232 | 0.294 | 0.241 |
| | ICL-All | 0.651 | 0.348 | 0.000 | 2.327 | 0.383 | 0.232 | 0.282 | 0.291 |
| | Vanilla SFT | 0.860 | 0.139 | 0.000 | 1.599 | 0.395 | 0.325 | 0.348 | 0.371 |
| | *ContextAgent* | 0.907 | 0.093 | 0.000 | 1.462 | 0.503 | 0.430 | 0.445 | 0.468 |

Table 13: Results for the Level-3 samples in ContextAgentBench.

| Model | Method | Proactive Predictions | | | | Tool Calling | | | |
|---|---|---|---|---|---|---|---|---|---|
| | | Acc-P$\uparrow$ | MD.$\downarrow$ | FD.$\downarrow$ | RMSE$\downarrow$ | Precision$\uparrow$ | Recall$\uparrow$ | F1-score$\uparrow$ | Acc-Args$\uparrow$ |
| GPT-4o | Proactive Agent | 0.714 | 0.285 | 0.000 | 2.604 | 0.523 | 0.158 | 0.239 | 0.253 |
| | Vanilla ICL | 0.857 | 0.142 | 0.000 | 1.817 | 0.732 | 0.332 | 0.434 | 0.466 |
| | CoT | 0.910 | 0.089 | 0.000 | 1.564 | 0.784 | 0.406 | 0.504 | 0.534 |
| | ICL-P | 0.928 | 0.071 | 0.000 | 1.463 | 0.794 | 0.357 | 0.467 | 0.598 |
| | ICL-All | 0.946 | 0.053 | 0.000 | 1.309 | 0.794 | 0.415 | 0.516 | 0.598 |
| GPT-3.5-Turbo | Proactive Agent | 1.000 | 0.000 | 0.000 | 0.845 | 0.517 | 0.149 | 0.231 | 0.068 |
| | Vanilla ICL | 0.982 | 0.017 | 0.000 | 1.069 | 0.702 | 0.244 | 0.351 | 0.377 |
| | CoT | 0.946 | 0.053 | 0.000 | 1.093 | 0.647 | 0.337 | 0.422 | 0.440 |
| | ICL-P | 1.000 | 0.000 | 0.000 | 0.790 | 0.741 | 0.305 | 0.408 | 0.408 |
| | ICL-All | 0.981 | 0.017 | 0.000 | 0.886 | 0.727 | 0.374 | 0.462 | 0.468 |
| Qwen2.5-72B-Ins | Proactive Agent | 0.928 | 0.071 | 0.000 | 1.797 | 0.669 | 0.237 | 0.343 | 0.144 |
| | Vanilla ICL | 0.964 | 0.035 | 0.000 | 1.043 | 0.738 | 0.407 | 0.503 | 0.482 |
| | CoT | 0.964 | 0.035 | 0.000 | 1.101 | 0.720 | 0.451 | 0.528 | 0.477 |
| | ICL-P | 1.000 | 0.000 | 0.000 | 0.790 | 0.770 | 0.422 | 0.518 | 0.504 |
| | ICL-All | 1.000 | 0.000 | 0.000 | 0.916 | 0.717 | 0.428 | 0.511 | 0.487 |
| Llama3.1-70B-Ins | Proactive Agent | 0.946 | 0.053 | 0.000 | 1.232 | 0.595 | 0.268 | 0.356 | 0.318 |
| | Vanilla ICL | 1.000 | 0.000 | 0.000 | 0.668 | 0.738 | 0.398 | 0.497 | 0.381 |
| | CoT | 1.000 | 0.000 | 0.000 | 0.801 | 0.758 | 0.407 | 0.497 | 0.392 |
| | ICL-P | 1.000 | 0.000 | 0.000 | 0.719 | 0.708 | 0.394 | 0.480 | 0.386 |
| | ICL-All | 1.000 | 0.000 | 0.000 | 0.707 | 0.668 | 0.410 | 0.482 | 0.460 |
| DeepSeek-R1-7B | Proactive Agent | 0.285 | 0.714 | 0.000 | 3.964 | 0.142 | 0.048 | 0.070 | 0.040 |
| | Vanilla ICL | 0.607 | 0.392 | 0.000 | 2.991 | 0.251 | 0.125 | 0.158 | 0.081 |
| | CoT | 0.517 | 0.482 | 0.000 | 3.348 | 0.238 | 0.105 | 0.137 | 0.111 |
| | ICL-P | 0.625 | 0.375 | 0.000 | 2.945 | 0.363 | 0.148 | 0.203 | 0.130 |
| | ICL-All | 0.500 | 0.500 | 0.000 | 3.313 | 0.279 | 0.117 | 0.158 | 0.098 |
| | Vanilla SFT | 0.946 | 0.053 | 0.000 | 1.093 | 0.714 | 0.467 | 0.542 | 0.400 |
| | *ContextAgent* | 1.000 | 0.000 | 0.000 | 0.755 | 0.721 | 0.444 | 0.529 | 0.455 |
| Qwen2.5-7B-Ins | Proactive Agent | 0.821 | 0.178 | 0.000 | 2.129 | 0.392 | 0.199 | 0.255 | 0.039 |
| | Vanilla ICL | 0.910 | 0.089 | 0.000 | 1.586 | 0.506 | 0.263 | 0.334 | 0.332 |
| | CoT | 0.892 | 0.107 | 0.000 | 1.742 | 0.425 | 0.196 | 0.258 | 0.271 |
| | ICL-P | 0.910 | 0.089 | 0.000 | 1.690 | 0.488 | 0.251 | 0.316 | 0.306 |
| | ICL-All | 0.910 | 0.089 | 0.000 | 1.679 | 0.410 | 0.184 | 0.246 | 0.305 |
| | Vanilla SFT | 0.946 | 0.053 | 0.000 | 1.157 | 0.731 | 0.504 | 0.576 | 0.417 |
| | *ContextAgent* | 0.964 | 0.035 | 0.000 | 0.972 | 0.672 | 0.489 | 0.549 | 0.465 |
| Llama3.1-8B-Ins | Proactive Agent | 1.000 | 0.000 | 0.000 | 0.731 | 0.538 | 0.177 | 0.260 | 0.090 |
| | Vanilla ICL | 0.732 | 0.267 | 0.000 | 2.228 | 0.526 | 0.174 | 0.257 | 0.235 |
| | CoT | 0.821 | 0.178 | 0.000 | 1.889 | 0.567 | 0.225 | 0.308 | 0.264 |
| | ICL-P | 0.625 | 0.375 | 0.000 | 2.741 | 0.508 | 0.153 | 0.233 | 0.245 |
| | ICL-All | 0.875 | 0.125 | 0.000 | 1.752 | 0.574 | 0.222 | 0.306 | 0.299 |
| | Vanilla SFT | 0.946 | 0.053 | 0.000 | 1.149 | 0.620 | 0.389 | 0.459 | 0.369 |
| | *ContextAgent* | 0.964 | 0.035 | 0.000 | 1.043 | 0.657 | 0.406 | 0.477 | 0.466 |

