# OpenReview forum: "ContextAgent: Context-Aware Proactive LLM Agents with Open-world Sensory Perceptions"
_NeurIPS.cc/2025/Conference — NeurIPS 2025 poster_

### Official Review · Reviewer_D7kT · 2025-06-08

**Clarity:** 2
**Significance:** 3
**Originality:** 3
**Rating:** 4
**Confidence:** 3

**Summary:**

This paper introduces ContextAgent, a context-aware proactive agent designed to operate under open-world sensory conditions. The authors propose two new benchmark, contextagentbenchmark and lite version, which combines LLM-assisted generation with human curation. The benchmark includes rich multimodal information such as egocentric video, audio, reasoning traces, and user personas. Experimental results demonstrate that ContextAgent achieves SOTA performance compared to existing LLMs on this benchmark.

However, the paper suffers from unclear descriptions in the task definition, task construction, and experimental comparisons, which significantly affects readability and makes it difficult to fully follow the methodology.

**Questions:**

1. In line 133, what exactly is the definition of $TR$?
2. Why does the vanilla SFT baseline not include persona information during training? This seems to create an imbalance in the comparison against ContextAgent.
3. What is the composition of the D_SFT training data? Please provide more details on the dataset size, construction. etc.
4. What does the persona data look like in practice? Can you provide concrete examples to clarify the structure and content of persona annotations?

**Ethical Concerns:**

["NO or VERY MINOR ethics concerns only"]

**Final Justification:**

The author's reply answered some of my questions, so I raised the score. However, I think the author's revisions need to be added to the main text to make the article more complete.

**Limitations:**

yes

**Quality:**

2

**Strengths And Weaknesses:**

Strengths:
1. The proposed ContextAgentBench dataset is well-motivated and addresses a practical need for evaluating proactive LLM agents in open-world, multimodal settings.
2. The authors conduct extensive experiments across 10 different LLMs, including both proprietary and open-source models.
3. The paper includes clear and informative figures that effectively illustrate the system architecture, data structure, and experimental results.

Weakness：
1. In Section 3.1 Task Definition and Task Construction, many variables are introduced but inconsistently notated (e.g., $P_S$  vs. PS) and defined out of order across paragraphs. Key terms such as the trigger threshold TR are underexplained, and the overall formulation lacks a clear separation between inputs, outputs, and constants—making the method harder to follow and reducing its interpretability.
2. Although the authors conduct extensive experiments, I find the experimental setup somewhat problematic. Specifically, ContextAgent is trained via supervised fine-tuning, yet it is compared against several training-free baselines (e.g., ICL variants), where the control over input information appears loosely related to the actual task setting (sft). Additionally, the vanilla SFT baseline does not include persona information as input, which results in an unfair comparison due to missing critical context.
3. The proposed agent does not demonstrate clear structural or architectural novelty. It primarily builds on existing LLMs and applies domain-specific supervised fine-tuning.
4. While the paper mentions that ContextAgent is fine-tuned using supervised learning, it does not provide sufficient details about the training data $D_SFT$.
5. The paper does not provide concrete examples from the dataset, nor does it release the dataset itself. As a result, it is unclear what the persona annotations look like, or how representative they are.

I may have missed certain details, and I am open to adjusting my score based on the authors’ response and clarifications.

---

> ### Author Rebuttal · Authors · 2025-07-27
>
> # W1 and Q1. Concern about the task formulation and notations.
> # Response:
> - We would like to clarify that this is a typo, which should be $\mathcal{P_S}$, i.e., the proactive score. We will correct this typo by changing $\mathcal{PS}$ to $\mathcal{P_S}$ and thoroughly proofread the entire paper to ensure consistency in notation and a logical order of variable definitions throughout.
> - $TR$ is the threshold for the proactive trigger, as defined in Sec.4.2. To improve the clarity, we use $\theta$ instead of $TR$ to represent this threshold. When the agent's predicted proactive score $\mathcal{P_S}$ exceeds $\theta$ (i.e., $\mathcal{P_S} > \theta$), the agent proactively provides suggestions to the user. Otherwise, ContextAgent avoids interrupting the user. In practice, this threshold can be manually adjusted to different levels by users based on their preferences for proactivity. For example, users who prefer highly sensitive proactive services, like for visual impairments, can lower the threshold to 2, while those needing less frequent assistance can raise it to 4. In our experiments, since we focus on evaluating common-sense sensitivity, the threshold is set to 3. We will add more descriptions of the trigger threshold $\theta$ in Sec.3.1.
> - To enhance the clarity of constants, we replace the threshold $TR$ with $\theta$, and use $\textbf{T}$ to represent the tool set that the agent can use. To better illustrate the inputs, outputs, and constants, we provide a table as follows:
> |Notation|Definition|Function|
> |-|-|-|
> |$\mathcal{S}$|Sensory perceptions|Inputs|
> |$\mathcal{P}$|Personas|Inputs|
> |$\mathcal{T}$|Thought traces|Outputs|
> |$\mathcal{P_S}$|Proactive score|Outputs|
> |$\mathcal{T_C}$|Tool chains|Outputs|
> |$\mathcal{R}$|Response|Outputs|
> |$\theta$|Proactive trigger threshold|Constants|
> |$\mathbf{T}$|Tool set|Constants|
>
> We hope this clarification can address your concerns. We will use clearer notations to enhance the overall formulation in Sec.3.1.
> # W2, W3 and Q2. Concerns about the experimental setups and the contribution of this work.
> # Response:
> - First, we would like to clarify that **our work is the first to study: context-aware proactive agents**. Existing studies on LLM agents primarily adopt a **reactive** paradigm, which requires explicit user instructions to activate their services. **In contrast, we first propose a new research problem aimed at leveraging the massive contexts surrounding humans, which can be captured through various wearable and mobile devices, to enhance the proactivity of LLM agents**.  The paradigm of context-aware proactive agent proposed in this work needs to map sensory contexts into intention understanding and tool calling, rather than relying on direct user language instructions. Effectively leveraging these contexts surrounding humans to develop more intelligent and **proactive agents** is a promising research direction. This advancement will not only **benefit the community** but also drive the **development of advanced, human-centric, and proactive AI assistants**.
> - Second, to address this new research problem, we propose a framework called ContextAgent, where we innovatively model this new task by incorporating sensory contexts, personas, and the reasoning traces distilled from advanced LLMs for enhanced proactive reasoning. Additionally, we introduce ContextAgentBench, the first benchmark for context-aware proactive agent tasks, designed for this new research problem. Each dataset sample includes multimodal sensory perceptions, sensory contexts, personas, reasoning traces (indicating whether and what proactive services are provided), proactive scores, tool chains, and final responses. To understand the rationale behind the need for proactive services and their potential benefits, we extract reasoning traces distilled from the latest reasoning LLMs. These reasoning traces not only benefit the training of smaller LLMs but also enhance the explainability of this new task. **We believe our pioneering work on context-aware proactive agents and the developed benchmark will greatly benefit the communities.**
> - Third, **to the best of our knowledge, no prior work can be directly used as a baseline for this new task**. The most closely related work is Proactive Agent, which focuses on enclosed computer usage scenarios and does not leverage open-world sensory contexts or perform tool calling. Besides Proactive Agent, ICL-based and vanilla SFT, **we add two baselines for comparison: SFT-Persona and SFT-PCE. Both methods use all the training data for SFT (including personas)**. Beyond vanilla SFT, SFT-Persona employs persona information as input. SFT-PCE further adopts the proactive-oriented context extraction method used by ContextAgent. Here are the results:
> Model|Setting|Acc-P|MD|FD|RMSE|Precision|Recall|F1|Acc-Args|
> |-|-|-|-|-|-|-|-|-|-|
> Qwen2.5-7B-Ins|SFT-Persona|77.4%|8.3%|14.3%|1.79|48.1%|49.5%|47.3%|37.4%|
> ||SFT-PCE|79.7%|6.6%|13.7%|1.72|59.3%|59.8%|57.2%|40.4%|
> ||ContextAgent|85.0%|5.0%|10.0%|1.40|66.7%|61.5%|62.4%|47.9%|
> Llama3.1-8B-Ins|SFT-Persona|73.4%|11.5%|15.1%|1.98|55.5% |58.2%|55.2%|35.3%|
> ||SFT-PCE| 78.6%|3.6%|17.9%|1.83|58.1%|56.3%|55.5%|41.8%|
> ||ContextAgent|83.0%|7.0%|10.0%|1.51|68.7%|63.7%|64.5%|48.6%|
> DeepSeek-R1-7B|SFT-Persona|76.2%|13.1%|10.7%|1.93|46.3%|46.5%|45.6%|22.8%|
> ||SFT-PCE|77.4%|16.7%|6.0%|1.81|54.6%|55.0%|53.5%|28.0%|
> ||ContextAgent|84.0%|5.0%|11.0%|1.51|67.8%|64.1%|64.8%|48.7%|
>
> ContextAgent employs both proactive-oriented sensory contexts, personas, and reasoning traces distilled from advanced LLMs for enhanced SFT, **consistently achieving higher performance than other personas and SFT-based baselines**. We hope this clarification accurately reflects the scope of our work and addresses your concerns.
> # W5 and Q4. Dataset details.
> # Response:
> Here are examples from our dataset, including sensory contexts, personas, thought traces, proactive scores, and tool calling. These personas enhance our benchmark's practicality, as individuals with varying personas may require different levels of proactivity, even experiencing the same sensor contexts. Reasoning traces (rationale behind the need for proactive services and their potential benefits) are distilled from the latest reasoning LLMs. They not only benefit the training of smaller LLMs but also enhance the explainability of this new task. We will release the code and the benchmark in the final version. We hope they can help you understand our dataset (we will update more examples in the appendix).
> ```json
> {
> "Context information":"Visual information shows the user is sitting at a dining table with a plate of food that includes pasta, grilled chicken, and a side of salad.",
> "Personas":[
> "The user has a history of managing Type 2 diabetes.",
> "The user is health-conscious and actively manages their diet to control blood sugar levels."
>  ],
> "Thoughts":"The user's personas show his glucose level is slightly elevated, and the user might appreciate dietary advice to help stabilize it. Providing this information could be helpful without being intrusive.",
> "Proactive score":4,
> "Tools": "[{"name":"get_health_data","desc":"Get health data from the user's smart device.","params":"None"},{"name":"wikipedia_search","desc":"Tool that searches the Wikipedia API.","params": {"query":"The effect of pasta, grilled chicken, and a side of salad on stabilizing blood sugar levels."}}]"
> "Response":"Your glucose level appears slightly elevated. To help stabilize it, you might consider increasing the portion of salad or reducing the amount of pasta. Grilled chicken is a great choice for protein, as it has minimal impact on blood sugar levels. Would you like me to provide additional dietary tips or help track your meal for better management?"
> }
> ```
> # W4 and Q3. Details about the training data.
> # Response:
> We construct $D_{SFT}=\{(\mathcal{X},\mathcal{T},\mathcal{Y})\}$, where inputs $\mathcal{X}$ include sensory context and personas, $\mathcal{T}$ is thought traces, and outputs $\mathcal{Y}$ include proactive scores and tool chains. Details can be seen in Sec.4.2. ContextAgentBench contains 1000 samples, with 60% of the samples used to build $D_{SFT}$. Here are examples from the SFT dataset.
> ```json
> {
> "Instruction":"You are a proactive LLM agent. You will receive both sensory and persona contexts. The sensory contexts are captured by the multi-modal sensors on the user's wearable devices. Persona is the personal information like identity and preference. Based on these contexts, you need to generate a thought and determine whether to initiate a proactive service for the user or not. When generating the response, consider whether you need to call external tools to complete the task. The following are the tool names and their arguments. Please only select tools from the specified tool sets and make sure to use the correct format for their arguments.## Tool set: {tool_set}",
> "input":"Sensory Context:Visual information shows the user is sitting at a dining table with a plate of food that includes pasta, grilled chicken, and a side of salad. Persona Context:The user has a history of managing Type 2 diabetes. The user is health-conscious and actively manages their diet to control blood sugar levels.",
> "output":"<think>The user's personas show his glucose level is slightly elevated, and the user might appreciate dietary advice to help stabilize it. Providing this information could be helpful without being intrusive.<\think> ## Proactive Predictions\n  "Proactive score":4  ## Tool Calling: [{"name":"get_health_data", "desc":"Get health data from the user's smart device.", "params":"None"},{"name":"wikipedia_search", "desc":"Tool that searches the Wikipedia API.","params":{"query":"The effect of pasta, grilled chicken, and a side of salad on stabilizing blood sugar levels."}}]"
> }
> ```
> We hope they can help you understand our SFT dataset. We will add more details about its construction in Sec.4.2.

---

> > ### Comment · Reviewer_D7kT · 2025-08-05
> >
> > Thank you for the detailed response, which addressed some of my concerns. However, after revisiting the paper, I could not find concrete details about the training dataset size in Section 4.2, which seems inconsistent with the information provided in the rebuttal. Given that the proposed method relies on supervised fine-tuning (SFT), I believe such details are essential and should be explicitly included in the paper. Additionally, the discussion around ICL-based baselines remains unconvincing.
> > After reviewing the comments from other reviewers, I also believe that several experimental results mentioned in the rebuttal should be incorporated into the main paper. Therefore, I think the paper still requires a major revision before it can be considered for acceptance.

---

> > > ### Author Response · Authors · 2025-08-05
> > >
> > > We sincerely appreciate your feedback and suggestions.
> > > - First, we would like to clarify that **we have provided the methodology of SFT dataset construction in Sec.4.2**, which **is consistent with our response to Q3** (details on D_SFT construction). **We will add the detailed description to the paper**, including the training size, examples of data samples, and comparisons with SFT-based baselines, as clarified in our rebuttal.
> > > - Further, several ICL-based approaches are used as baselines because employing SFT for comparison is infeasible for many proprietary LLMs (e.g., GPT-4o), and ICL remains a widely-used training-free approach for domain-specific tasks. Tab.4 in our main paper demonstrates that, for all ICL variants, incorporating reasoning traces and personas provides significant benefits for the context-aware proactive agent task. Since ICL relies on demonstrations within prompts, its limited generalizability leads to far lower performance on small-sized LLMs compared to ContextAgent. However, for advanced proprietary LLMs, ContextAgent achieves comparable and slightly lower performance.
> > > - Given that our work is the first to explore proactive agents in context-aware settings, we believe that our proposed framework and benchmark can benefit the community in the development of advanced, human-centric, and proactive AI assistants.
> > >
> > > We will add these clarifications and detailed description to the final version. We look forward to continuing the discussion and addressing any remaining points to ensure that all concerns are fully resolved.

---

> > > ### Author Response · Authors · 2025-08-09
> > >
> > > We truly appreciate the time and effort you've dedicated to reviewing our paper and sharing your insights.
> > >
> > > As the rebuttal deadline approaches, we would appreciate knowing whether our explanations have helped address your concerns. We would like to further clarify two points:
> > >
> > > - First, we would like to clarify that **we have provided a detailed analysis of the ICL-based baselines in Appendix E (lines 558–568) in our original paper.** In this section, we provide a detailed analysis of the performance of ICL variant baselines, including the significant impact of reasoning traces, the influence of personas, and their effects on different sizes of LLMs (e.g., 7B and 70B). We hope this clarification can address your concerns!
> > > - Second, we would like to clarify that we described the SFT dataset construction process in Sec.4.2 of our original paper. **This is consistent with our response in Rebuttal**. We will add the description of the training data size to the camera-ready version. We hope this clarification can address your concerns!
> > >
> > > We would also clarify that our work is the first to study a context-aware proactive agent, which harnesses the massive sensory contexts surrounding humans to enhance the proactivity of LLM agents. We believe this pioneering study, along with the developed benchmark, has the potential to drive the development of advanced, human-centric, and proactive AI assistants, benefiting research communities.
> > >
> > > We’re eager to continue the conversation and would love to hear any additional insights or suggestions you might have. Thank you once again for your valuable feedback. We look forward to your response!

---

### Official Review · Reviewer_HAmH · 2025-07-01

**Clarity:** 3
**Significance:** 2
**Originality:** 3
**Rating:** 4
**Confidence:** 3

**Summary:**

This paper proposed a benchmark for evaluating context-aware proactive LLM agents and an author-designed agent as baseline model. The baseline agent is a context-aware proactive agent that fuses rich sensory data from video, audio with a user’s historical persona to infer intentions and decide when help is needed, then autonomously invokes the right tools to assist unobtrusively. The proposed ContextAgentBench contain nine everyday scenarios and twenty tools. On this benchmark, ContextAgent surpasses prior baselines.

**Questions:**

Q1. Why don't you use VLM as the main agent directly?

Q2. About W2. Discuss the situation when some external tools are not aviliable, can your agent still work?

Q3: ContextAgent is a multi-tool agent in this stage, why don't you consider code agent for this problem.

**Ethical Concerns:**

["NO or VERY MINOR ethics concerns only"]

**Final Justification:**

Based on authors' response and our discussion, I update my score to borderline.

**Limitations:**

They discuss limitations in the supp.

**Paper Formatting Concerns:**

None.

**Quality:**

3

**Strengths And Weaknesses:**

S1. The authors proposed a personalized agentic framework, which match well with the community's interests.

S2. They tested the ContextAgent with open-sourced LLMs (including small LLMs), which is useful for re-implementation.

S3. The main idea is simple and clear and also contain OOD experiment.

W1: Comparison might be unfair in Table 1 because some of other ideas are zero-shot method. Only Vanilla SFT and your proposed idea utilize the training data.

W2: Do not contain missing sensor situation. e.g., the input data of some sensor is not aviliable, what might be the context output?

W3: For 5.4 Out-of-Domain Evaluation, it would be good to compare your method with the latest GPT/Claude series like o3.

---

> ### Author Rebuttal · Authors · 2025-07-27
>
> # W1:  Concern about the tested baselines in Table 1.
> # Response:
> - First, we would like to clarify that **our work is the first to study: how to build and evaluate a context-aware proactive agent**. Existing studies on LLM agents primarily adopt a **reactive** paradigm, which requires explicit user instructions to activate their services. In contrast, **we first propose a new research problem aimed at leveraging the massive contexts surrounding humans, which can be captured through various wearable and mobile devices, to enhance the proactivity of LLM agents**.  The paradigm of context-aware proactive agent proposed in this work needs to map sensory contexts into intention understanding and tool calling, rather than relying on direct user language instructions. This advancement will not only **benefit the community** but also drive the **development of advanced, human-centric, and proactive AI assistants**.
> - Second, **to the best of our knowledge, no prior work can be directly used as a baseline for this new task**. The most closely related work is Proactive Agent, which focuses on enclosed computer usage scenarios and does not leverage open-world sensory contexts or perform tool calling. To ensure the fairness of comparison, besides Proactive Agent, ICL-based and vanilla SFT, **we add two baselines for comparison: SFT-Persona and SFT-PCE. Both methods use all the training data for SFT**. Beyond vanilla SFT, SFT-Persona employs persona information as input. SFT-PCE further adopts the proactive-oriented context extraction method used by ContextAgent. Here are the results:
> Model|Setting|Acc-P|MD|FD|RMSE|Precision|Recall|F1|Acc-Args|
> |-|-|-|-|-|-|-|-|-|-|
> Qwen2.5-7B-Ins|SFT-Persona|77.4%|8.3%| 14.3%|1.79|48.1%|49.5%| 47.3%| 37.4%|
> ||SFT-PCE|79.7%|6.6%|13.7%|1.72|59.3%|59.8%|57.2%|40.4%|
> ||ContextAgent|85.0%| 5.0%|10.0%|1.40|66.7%|61.5%|62.4%|47.9%|
> Llama3.1-8B-Ins|SFT-Persona|73.4%|11.5%|15.1%|1.98|55.5%|58.2%|55.2%| 35.3%|
> ||SFT-PCE|78.6%|3.6%|17.9%|1.83| 58.1%|56.3%|55.5% |41.8%|
> ||ContextAgent|83.0%|7.0%|10.0%|1.51|68.7%|63.7%|64.5%|48.6%|
> DeepSeek-R1-7B|SFT-Persona|76.2%|13.1%|10.7%|1.93|46.3%|46.5%|45.6%|22.8%|
> ||SFT-PCE|77.4%|16.7%|6.0%|1.81|54.6%|55.0%|53.5%| 28.0%|
> ||ContextAgent|84.0%|5.0%|11.0%|1.51|67.8%|64.1%|64.8%|48.7%|
>
> ContextAgent employs both proactive-oriented sensory contexts, personas, and reasoning traces distilled from advanced LLMs for enhanced SFT, **consistently achieving higher performance than other personas and SFT-based baselines**. We hope this clarification accurately reflects the scope of our work and addresses your concerns.
> # W2: Performance on missing sensor situation.
> # Response:
> Here is an example where the input data from certain sensors is unavailable. Consider a context that originally includes both vision and audio: *"The person is engaging in a conversation with a colleague in an office. The content is: Hi, how about going to City A this weekend? I want to go to the beach."* When the audio sensor is missing, the context output is: *"The person is engaging in a conversation with a colleague in an office."* When the visual sensor is missing, the context is reduced to: *"Hi, how about going to City A this weekend? I want to go to the beach."*
>
> To clearly show the sensor data's impact, we **added new experiments on missing different sensor modalities**. Results show that missing either the vision or audio leads to performance degradation for both proactive prediction and tool calling. The impact of missing vision is more significant than that of missing the audio. Here are the results.
> || Acc-P|MD|FD|RMSE|Precision|Recall|F1|Acc-Args|
> |-|-|-|-|-|-|-|-|-|
> |Full|88.8%|2.7%|8.5%|1.31|67.6%|64.8%|64.7%|46.8%|
> |Miss vision|70.9%|12.9%|16.3%|2.07|42.4%|42.2%|41.4%|16.3%|
> |Miss audio|72.0%|17.3%|10.6%|2.02|50.1%|50.1%|49.3%|21.2%|
> # W3: Comparison with the latest reasoning LLMs.
> # Response:
> We **added new experiments using advanced reasoning LLMs** in both zero-shot (ZS) and ICL settings. Results show that among these reasoning models, Claude Opus 4 achieves the highest scores in both proactive predictions and tool calling. In addition, ContextAgent shows comparable performance to advanced reasoning models for Acc-P, while its tool-calling performance, such as F1, is lower compared to Claude Opus 4. It may be due to the gap in model size, as ContextAgent uses a 7B-scale LLM.
> |Model|Acc-P|MD|FD| RMSE|Precision|Recall|F1| Acc-Args|
> |-|-|-|-|-|-|-|-|-|
> |o4-mini (ZS)|70.5%|1.7%|27.6%|1.57|63.1%|53.5%|55.9%|42.4%|
> |o4-mini (ICL)|86.1%|3.4%|10.3%|1.24|72.6%|66.8%|68.2%|53.8%|
> |GPT-o3 (ZS)|83.0%|9.0%|7.9%|1.34|71.6%|64.9%|66.7%|39.7%|
> |GPT-o3 (ICL)|86.8%|6.9%|6.2%|1.10|75.5%|69.7%|71.1%|56.3%|
> |Claude Sonnet 4 (ZS)|83.7%|11.0%|5.1%|1.34|75.7%|69.0%|70.8%|23.7%|
> |Claude Sonnet 4 (ICL)|91.3%|6.9%|1.7%|1.01|77.5%|79.9%|77.3%|48.0%|
> |Claude Opus 4 (ZS)|83.0%|7.2%|9.6%|1.27|73.8%|66.0%|68.0%| 24.4%|
> |Claude Opus 4 (ICL)|93.7%|4.1%|2.0%|0.86|81.4%|85.6%|81.8%| 56.8%|
> |ContextAgent (Qwen2.5-7B-Ins)|90.9%|2.0%|7.0%|1.17|71.1%|69.9%|68.9%|51.6%|
> # Q1. Performance when using VLM as the main agent.
> # Response:
> First, we would like to clarify that we chose LLMs over VLMs because they use a unified text format to represent contexts across all sensor modalities, making them more scalable to new types of sensors. This modular design enhances the framework’s scalability, enabling seamless integration with various sensor modalities (e.g., IMU, video, sound) and emerging MLLMs. However, the ContextAgent framework also supports using other MLLMs as the main agent, such as VLM. We **added new experiments using VLMs as the main agent** in ContextAgent, while keeping all other components of the framework exactly the same. We tested various VLMs across different model sizes. Here are the results.
> |Model|Acc-P|MD|FD|RMSE|Precision|Recall|F1|Acc-Args|
> |-|-|-|-|-|-|-|-|-|
> |Qwen2.5-VL-7B-ICL|64.2%|8.3%|27.3%|2.18|35.9%|32.5%|32.3%|26.8%|
> |InternVL3-8B-ICL|60.7%|1.1%|38.1%|2.02|21.8%|24.3%|21.6%|19.4%|
> |LLaVA1.5-7B|69.0%|27.3%|3.5%|2.16|54.3%|54.0%|53.3%|29.7%|
> |Qwen2.5-VL-3B |83.3%|10.7%|5.9%|1.58|60.3%|59.5% |58.6%|36.7%|
> |Qwen2.5-VL-7B|84.5%|10.7%|4.7%|1.56|71.0%|68.0%|68.4%|51.1%|
> |InternVL3-2B|85.7%|14.2%|0.0%|1.44|67.8%|68.3%|67.1%|45.8%|
> |InternVL3-8B|84.5%|10.7%|4.7%|1.42|69.4%|71.0%|68.7%|54.3%|
> |Qwen2-VL-2B|75.0%|20.2%|4.7%|1.93|54.9%|55.1%|54.2%|36.0%|
> |Qwen2-VL-7B|80.9%|13.1%|5.9%|1.55|66.5%|67.0%|65.5%|47.0%|
>
> We conducted VLM experiments under both ICL and SFT settings. Results show that using VLM as the main agent perform significantly worse than LLM-based agents in ICL settings. This is because providing examples of the entire process in the prompt is more challenging for VLMs to understand compared to using a two-stage pipeline. However, in SFT settings, VLM as the main agent performs comparably to LLM-based agents. We will add the results to the appendix.
> # Q2. Discussion on external tools unavailable.
> # Response:
> We would like to clarify that ContextAgent can still work when certain tools are unavailable. Here are two examples.
> ```json
> # Example 1:
> {
> "Contexts":"The user is browsing wireless earbuds in the electronics section.",
> "Thoughts":"The user might be interested in wireless earbuds. Providing price comparisons and review information could help with their purchase decision.",
> "Proactive score":5,
> "Tools":[{"name":"get_online_product_price","desc":"Get the price of a product from an online store such as Amazon or eBay", "params":{"product_name":"wireless earbuds"}},{"name":"search_rednote","desc":"A platform where people share tips on travel, fitness..","params": {"query":"wireless earbuds reviews and recommendations"}}]
> "Response":"It looks like you're exploring wireless earbuds. Would you like help comparing prices and reviews to find the best deal?"
> }
> # Example 2:
> {
> "Contexts":"The user is sitting at a dining table with a plate of food that includes pasta, grilled chicken, and a side of salad.",
> "Personas":[
> "The user has a history of managing Type 2 diabetes.",
> "The user is health-conscious and actively manages their diet to control blood sugar levels."
>  ],
> "Thoughts":"The user is managing Type 2 diabetes and is health-conscious, providing nutritional insights about their meal could support better blood sugar control and dietary decisions.",
> "Proactive score":4,
> "Tools":"[{"name":"get_health_data","desc":"Get health data from the user's smart device.","params":"None"},{"name":"wikipedia_search", "desc":"Tool that searches the Wikipedia API.", "params": {"query":"The effect of pasta, grilled chicken, and a side of salad on stabilizing blood sugar levels."}}]"
> "Response":"Your meal of pasta, grilled chicken, and salad is a solid choice. To support Type 2 diabetes management, control pasta portions to avoid blood sugar spikes, while the chicken’s protein and salad’s fiber help slow carbohydrate absorption."
> }
> ```
> Results show that ContextAgent can still reason on the current contexts and identify the need for proactive assistance. When tools are unavailable, ContextAgent can either proactively offer users helpful suggestions and ask if they would like to use certain tools, or, in some cases, rely on LLM's intrinsic knowledge to assist users.
> # Q3. Discussion on the code agent.
> # Response:
> First, we would like to clarify that ContextAgent and its benchmark are implemented using an API-based framework, a widely adopted paradigm in prior research. However, ContextAgent is not limited to an API-based approach and can be extended to support other implementation methods, such as code-based frameworks. Many existing works on LLM agents primarily focus on pipeline, reasoning, or implementations (e.g., API-based, code-based, or MCP-based), but they are within a reactive paradigm. However, our work is **orthogonal** to theirs. Unlike existing reactive LLM agents that rely on explicit user instructions, we focus on designing context-aware proactive agents that leverage rich sensory data.

---

> > ### Author Response · Authors · 2025-08-08
> > **Kindly Request for Your Feedback Before Rebuttal Deadline**
> >
> > We truly appreciate the time and effort you've dedicated to reviewing our paper and sharing your insights.
> >
> > As the rebuttal deadline approaches, we would appreciate knowing whether our explanations have helped address your concerns regarding the performance of more advanced reasoning models and VLMs, as well as the performance in special situations. We would also clarify that our work is the first to study a context-aware proactive agent, which harnesses the massive sensory contexts surrounding humans to enhance the proactivity of LLM agents. We believe this pioneering study, along with the developed benchmark, has the potential to drive the development of advanced, human-centric, and proactive AI assistants, benefiting research communities.
> >
> > We’re eager to continue the conversation and would love to hear any additional insights or suggestions you might have. Thank you once again for your valuable feedback. We look forward to your response!

---

> > ### Comment · Reviewer_HAmH · 2025-08-08
> >
> > Thank you for your response. I appreciate your experiment on VLM as the main agent. This could be something you can do as a follow-up because well-finetuned VLM can understand egocentric videos. I am considering adjusting my final score to Borderline Accept during the discussion stage (will try to reach consensus with all reviewers). However, I still find it difficult to conclude that the current workflow is truly human-centric and proactive.
> >
> > Regarding the cases where external tools are unavailable, do you have any quantitative results? As far as I know, tool availability is a well-recognized challenge in many recent agentic AI papers. Presenting these results would help demonstrate that the model’s conclusions are not overly dependent on one or two dominant tools or sensors.

---

> ### Author Response · Authors · 2025-08-06
> **Kindly Requesting for You Feedback Before Rebuttal Deadline**
>
> Dear Reviewer,
>
> Thank you for your detailed review of our work. We sincerely appreciate your recognition of the concept of context-aware proactive agents and its alignment with the community's interests.
>
> We have carefully addressed each of your comments and believe further discussion could enhance our work. We would appreciate knowing whether our explanations have helped resolve your concerns regarding the performance of more advanced reasoning models and VLMs, as well as the performance in special situations. If there are still unresolved issues, we kindly invite your feedback. We are now ready to address any remaining misunderstandings or confusion you may have.
>
> With the rebuttal deadline approaching, we would be grateful for any additional suggestions, concerns, or comments at your earliest convenience. We are eager to continue the discussion and clarify any unsolved points to ensure all matters are thoroughly addressed.
>
> Thank you once again for your time and thoughtful consideration.

---

> ### Author Response · Authors · 2025-08-09
> **Response to Reviewer**
>
> We sincerely appreciate that our clarifications and experiments have addressed your concerns regarding the VLM agents.
>
> # Response to Follow-up Q1.
>
> - **Proactivity**. First, we would like to clarify that **ContextAgent leverages the rich sensory contexts surrounding humans as its system inputs, rather than relying on explicit user instructions**. These inputs are derived from user-worn and mobile devices, such as smart glasses (ego-centric video) and earphones (audio). ContextAgent maps sensory contexts into intention understanding and tool calling, which is different from existing reactive LLM agents that rely on user explicit instructions. Since ContextAgent does not rely on explicit user instructions to initiate its services, **it aligns with the definitions of proactive systems in previous studies [1].**
> - **Human-centric**. We would like to clarify that both the inputs and outputs of ContextAgent are highly relevant to humans. First, the **inputs** are derived from the rich sensory contexts surrounding humans, including ego-centric video and audio captured through **users' wearable and mobile devices**. Second, ContextAgent integrates both sensory contexts and personas for proactive predictions. The personas, reflecting an individual’s preferences and identity, further align with the design goal of a human-centric agent. Third, the **outputs** are proactive suggestions such as *"The weather forecast predicts heavy rain and thunderstorms with a temperature of 26°C this weekend, making it less ideal for outdoor team-building. Additionally, you have a scheduling conflict. It may be best to reschedule for a time with better weather and a clear schedule."* They can be **highly valuable in helping humans make informed decisions** in their daily lives.
>
> [1] Proactive Agent: Shifting LLM Agents from Reactive Responses to Active Assistance, ICLR'25.
>
>
>
> # Response to Follow-up Q2.
>
> - First, we would like to clarify that **the primary focus of ContextAgent is to develop a context-aware proactive agent** that leverages rich sensory contexts surrounding humans for proactive predictions and user-intended tool calling. Recent studies investigated LLM agents in tool-unavailable situations [2]. However, these agents operate under the reactive paradigm, which is **orthogonal to our work and out of the scope** of our research. In fact, ContextAgent can also integrate these latest approaches designed for handling tool-unavailable situations to further enhance its capabilities.
>
> - Second, although handling tool-unavailable situations is not the focus of our work, we have provided examples showcasing ContextAgent in such scenarios in our rebuttal (as shown in the example provided in our previous response to Q2). The results indicate that **ContextAgent is not overly reliant on one or two dominant tools** when generating proactive responses. For example, ContextAgent can either **proactively offer users helpful suggestions and ask if they would like to use certain tools, or, in some cases, rely on LLM's intrinsic knowledge to assist users**. For the quantitative results, we would like to clarify that the output of ContextAgent both contains proactive predictions, tool planning, and the final responses.
>
>   - For the outputs of proactive predictions and tool planning, tool-unavailable situations will not impact their scores.
>
>   - For the final proactive responses, we compare the semantic similarity of ContextAgent's outputs to the ground truth under two conditions: with all tools available and with one tool randomly made unavailable. The results show a 1.3% decrease in BERTScore (from 93.5% to 92.2%), indicating that ContextAgent does not overly rely on dominant tools when generating proactive responses.
>
>
>
> [2] Can Tool-augmented Large Language Models be Aware of Incomplete Conditions, arXiv.
>
>
>
> We would greatly appreciate knowing whether our explanations helped address your concerns. Your support and encouragement are invaluable to us! Thank you once again for your time and thoughtful consideration.

---

> > ### Comment · Reviewer_HAmH · 2025-08-09
> >
> > Thanks for your reply. I will finalize my score during the discussion phase.

---

> > > ### Author Response · Authors · 2025-08-09
> > >
> > > We are pleased to have addressed your concerns and sincerely appreciate the time and effort you have spent reviewing our paper. Your support and encouragement are invaluable to us!

---

### Official Review · Reviewer_EoSu · 2025-07-03

**Clarity:** 3
**Significance:** 4
**Originality:** 4
**Rating:** 5
**Confidence:** 4

**Summary:**

This paper introduces a context-aware proactive LLM agent, ContextAgent, along with a benchmark, ContextAgentBench, to evaluate context-aware proactive LLM agents. While prior work have created proactive LLM agents leverage observations from computer interfaces or repeatedly reask questions. ContextAgent leverages sensory information (from wearable devices) and makes uses of user personas to proactively call tools and provide users information. ContextAgent is evaluated using ContextAgentBench by the accuracy of proactive predictions and the tools called to provide proactive feedback.

**Questions:**

- What do proactive task with a score of 2, 3, and 4 look like?
  - How were these threshold of proactivity decided for the tasks? In Figure 8 some tasks involve a user working on a computer and the proactiveness score is 1 but I could imagine the user being stuck with an error and requiring some proactive assistance. Similarly for the case where the user is walking by some dried out plants which could prompt looking for water fountains nearby to water the plants.
- Is this benchmark already saturated given that ContextAgent achieves nearly 90~ accuracy on proactive predictions? What are failure modes and insights in closing the gap?
- Similar question for tool calling except that it seems there is room for improvement here.
- For ProactiveBench, why is it necessary to leverage contextual information from wearable devices on computer tasks? For a human using a computer, wouldn't a wearable device need to focus on the task at hand?
  - If this is the case, would it be possible to evaluate ContextAgent on ProactiveBench? Essentially the sensory information is distilled through the desktop UI and the persona information is distilled into the task instruction

**Ethical Concerns:**

["NO or VERY MINOR ethics concerns only"]

**Final Justification:**

The authors have addressed my main concern regarding benchmark saturation and minor clarifications on the proactiveness scoring and why evaluating ProactiveBench is out of scope for the paper. I have increased my rating and confidence score since there is still a significant performance gap to close on the Acc-Args metrics that cannot trivially be solved through ICL.

**Limitations:**

There is a discussion to be had on being context-aware for dangerous situations where the tool calls can contact emergency services for scenarios like (1) witnessing crimes or (2) health-related problems (e.g. fall detection on Apple watches is mentioned in the introduction).

**Quality:**

4

**Strengths And Weaknesses:**

**Quality:** Extensive experiments and ablations are conducted to demonstrate the usefulness of ContextAgent making use of sensory information and persona data.

**Clarity:** The paper is fairly easy to understand but there are some improvements that can be made in clarity in the Task Definition section (some terms like TR are mentioned but not defined until later sections). It is also not obvious what a proactive score of 2-4 means.

**Significance:** This is an important problem to study and the paper provides both a solution and benchmark for evaluation.

**Originality:** This paper proposes the first benchmark for context-aware proactive LLM agents.

---

> ### Author Rebuttal · Authors · 2025-07-27
>
> # Q1-1. Details about the dataset.
> # Response:
> First, we would like to clarify the rationale behind the annotation of proactive scores in our dataset: 5 means a clear need for proactive services, 1 indicates no need, 3 is neutral, and 2 and 4 represent slight tendencies. Below are three examples whose proactive scores are 2, 3, and 4 in our dataset, respectively. We hope they can help you understand our dataset (we will also update more examples in the appendix).
> ```json
> # Proactive score of 4
> {
> "Context information": "Visual information shows the user is sitting at a dining table with a plate of food that includes pasta, grilled chicken, and a side of salad.",
> "Personas": [
> "The user has a history of managing Type 2 diabetes.",
> "The user is health-conscious and actively manages their diet to control blood sugar levels."
>  ],
> "Thoughts": "The user's personas show his glucose level is slightly elevated, and the user might appreciate dietary advice to help stabilize it. Providing this information could be helpful without being intrusive.",
> "Proactive score": 4,
> "Tools": "[{"name": "get_health_data", "desc": "Get health data from the user's smart device.", "params": "None"},{"name": "wikipedia_search", "desc": "Tool that searches the Wikipedia API. Useful for when you need to answer general questions about people, places, companies, facts, historical events, or other subjects.", "params": {"query": "The effect of pasta, grilled chicken, and a side of salad on stabilizing blood sugar levels."}}]"
> "Response": "Your glucose level appears slightly elevated. To help stabilize it, you might consider increasing the portion of salad or reducing the amount of pasta. Grilled chicken is a great choice for protein, as it has minimal impact on blood sugar levels. Would you like me to provide additional dietary tips or help track your meal for better management?"
> }
>
> # Proactive score of 3
> {
> "Context information": "Visual information suggests that the user is looking in the mirror while wearing summer clothes, such as short sleeves and shorts.",
> "Personas": ["A university psychology student who is currently studying creativity and personality."],
> "Thoughts": "The user is looking in the mirror and may be preparing to go out, wearing minimal clothing such as short sleeves and shorts. I need to check the weather for the user's location. If the temperature is expected to drop or if rain is forecasted for the day, I can proactively remind the user to dress warmly to avoid catching a cold.",
> "Proactive score": 3,
> "Tools": "[{"name": "get_current_gps_coordinates", "desc": "Get the current GPS coordinates of the user", "params": "None"},{"name": "get_city_weather", "desc": "Get the weather for a specified city at a given time.", "params": {"city": "results of get_current_gps_coordinates", "time":"now"}}]"
> "Response": "I checked the weather forecast, and the local temperature is expected to drop sharply after 6 PM. Considering that you are only wearing short-sleeved shirts and shorts, it is recommended that you add more layers."
> }
>
> # Proactive score of 2
> {
> "Context information": "Visual information suggests the user is in a gym, lifting weights.",
> "Personas": ["A computer science major interested in developing software for audio manipulation and enhancement."],
> "Thoughts": "The user might need guidance on proper weightlifting techniques or reminders to take breaks between sets. However, without detailed persona information, such as their fitness level, I would rate this proactive behavior at 2 points. This rating indicates some potential for proactivity, but the necessity is not very high. After collecting more detailed persona information, I could provide more accurate proactive services.",
> "Proactive score": 2,
> "Tools": "None",
> "Response": "None"
> }
> ```
> # Q1-2. Discussion on the threshold for proactivity and examples in Fig.8.
> # Response:
> - First, we consistently set the proactivity threshold to 3 across all methods in our experiments. However, in practice, this threshold can be manually adjusted to different levels by users based on their preferences for proactivity. For example, users who prefer highly sensitive proactive services, like for visual impairments, can lower the threshold to 2, while those needing less frequent assistance can raise it to 4. However, in our experiments, we focus solely on evaluating common-sense sensitivity, so the threshold is set to 3 for all experiments. In addition, we will add the definition of the proactivity threshold in Section 3.1 to improve the clarity.
> - Second, we would like to clarify that the proactivity score in working scenarios is not always fixed (i.e., neither as low as 1 nor as high as 5) and varies depending on the specific context. In Fig. 8, the proactive score is 1 because the context (*a user typing on a keyboard while working on a computer in an office*) does not suggest a need for proactivity. However, the score may be higher in scenarios where the user appears confused.
> - Third, regarding the example of watering plants in Fig. 8. In this work, we model the context-aware proactive agent task as leveraging both sensory contexts and personas for reasoning. **This means that even with identical sensory contexts, different personas in the samples may result in varying proactive scores and tool usage**.
> For example, in our developed ContextAgentBench, two samples share the same sensory context: *"The user is in a gym, standing in front of a treadmill."* However, they differ in personas: *"The user is new to fitness and does not exercise regularly,"* and *"The user exercises frequently."* For the first sample, the annotated output is: *"Would you like a personalized treadmill workout plan? I can also suggest a motivational playlist to keep you energized. Just let me know!"* In contrast, for the second sample, the proactive score is 1, indicating no need to disturb the user.
> # Q2 and Q3. Discussion on the potential saturation of the benchmark and strategies to close the performance gap.
> # Response:
> - First, our experimental results in Tab.4~6 show that increasing the model size of LLMs can lead to obvious performance improvement on both proactive predictions and tool calling.
> - Additionally, **we conducted new experiments using the latest reasoning LLMs** (e.g., Claude and GPT-o3 series) in both zero-shot (ZS) and in-context learning (ICL) settings. Results show that advanced reasoning LLMs like Claude Opus 4-ICL can lead to improvements in Acc-P and F1 scores. This shows that enhancing the reasoning capabilities of LLMs has the potential to close this gap.
> Here are the results:
> |  | Acc-P| MD | FD| RMSE    | Precision | Recall   | F1 | Acc-Args |
> |-|-|--|-|--|--|--|-|-|
> | o4-mini (ZS)| 70.5%   | 1.73% | 27.6%  | 1.57 | 63.1% | 53.5%  | 55.9% | 42.4% |
> | o4-mini (ICL) | 86.1%   | 3.4% | 10.3%  | 1.24 | 72.6% | 66.8%  | 68.2% | 53.8% |
> | GPT-o3 (ZS) | 83.0%   | 9.0% | 7.9% | 1.34 | 71.6%  | 64.9%  | 66.7% | 39.7% |
> | GPT-o3 (ICL)  | 86.8%   | 6.9%| 6.2% | 1.10 | 75.5% | 69.7%  | 71.1% | 56.3% |
> | Claude Sonnet 4 (ZS)| 83.7% | 11.0%| 5.1% | 1.34  | 75.7%  | 69.0% | 70.8%  | 23.7% |
> | Claude Sonnet 4 (ICL) | 91.3% | 6.9% | 1.7% | 1.01 | 77.5% | 79.9%| 77.3% | 48.0% |
> | Claude Opus 4 (ZS) | 83.0%| 7.2% | 9.6% | 1.27| 73.8% | 66.0% | 68.0% | 24.4% |
> | Claude Opus 4 (ICL) | 93.7% | 4.1% | 2.0% | 0.86 | 81.4% | 85.6% | 81.8% | 56.8% |
> # Q4. Discussion on the necessity of leveraging contextual information from wearable devices for computer tasks.
> # Response:
> - First, **opportunities for proactive assistance in computer usage or working scenarios can arise from many different perspectives**. Previous work on Proactive Agents and their proposed benchmark primarily focuses on scenarios related to computer usage, using inputs such as mouse clicks and keyboard types. However, **our work primarily focuses on leveraging rich sensory contexts surrounding humans**, such as vision, audio, and IMU data, rather than relying on computer inputs. These richer contexts surrounding humans, captured through various wearable and mobile devices, create opportunities to enhance LLM agents' proactivity. For example, from a **health perspective**, users can be proactively alerted with health-related suggestions, such as reminders to take breaks or engage in physical activity after sitting for extended periods, based on data from IMU sensory contexts. Additionally, in a "**computer usage scenario during an in-person meeting**," wearable devices can offer work-related tips, such as using audio sensory data to identify potential scheduling conflicts during discussions about future plans.
> - Second, the wearable devices in our benchmark are not designed to detect computer inputs, such as mouse clicks or keyboard strokes. Instead, they are used to capture sensory contexts surrounding humans, enabling more opportunities for proactive assistance during computer usage, such as health alerts or work-related tips during meetings. Furthermore, our proposed ContextAgent and benchmark enable the development of a proactive agent that extends beyond computer-centric scenarios, offering a wider range of opportunities for proactive assistance.
>
> # L1. Discussion on contacting emergency services.
> # Response:
> We will add more discussion on context awareness in dangerous situations, where the tool calls can contact emergency services for specific scenarios.

---

> ### Author Response · Authors · 2025-08-06
> **Kindly Requesting for You Feedback Before Rebuttal Deadline**
>
> Dear Reviewer,
>
> Thank you for your detailed review of our work.  We sincerely appreciate your recognition of the significance of our research on context-aware proactive agents, as well as your acknowledgment of the proposed solution, benchmark, and extensive experimental evaluation.
>
> We have carefully provided clarifications to each of your comments and believe further discussion could enhance our work. We would appreciate knowing whether our explanations have clarified your concerns regarding the dataset and the potential saturation. If there are still unresolved issues, we kindly invite your feedback. We are now ready to address any remaining misunderstandings or confusion you may have.
>
> With the rebuttal deadline approaching, we would be grateful for any additional suggestions, concerns, or comments at your earliest convenience. We are eager to continue the discussion and clarify any unsolved points to ensure all matters are thoroughly addressed.
>
> Thank you once again for your time and thoughtful consideration.

---

> ### Comment · Reviewer_EoSu · 2025-08-07
>
> Thank you for the detailed response. I have a better understanding of the nuances between scores 2, 3 and 4, how these thresholds were set, and how ProactiveBench is out of scope of this paper.
>
> I would like to better understand the point on saturation before considering raising my score. Before ContextAgent's Acc-P was 0.894 for proactive predictions and the F1-score for tool calling was 0.645 and I was concerned about saturation and the new results on Claude Opus 4 with ICL brought those scores up to 93.7% and 81.8% which deepened my concern. With the performance improvement on those metrics from zero-shot to ICL being 10.7% and 13.8% respectively, it seems like the remaining gap could be closed with better examples in the prompt. Is this intuition correct or do you feel as though there is a bigger challenge that needs to be addressed that isn't solely prompting based?
>
> In addition, the metric that seems to be lowest that could use improvement is Acc-Args. Can you please give some examples that detail why this is difficult? Particularly,
>
> >  If an argument of any tool is incorrect, the entire sample is considered incorrect
>
> Is this essentially a metric for whether the tool calls the agent produces match exactly with the tools in the ground-truth set? What if there was partial-credit assigned for matching some but not all; how much gap exists there for the agent to be aligned with the tool calls the benchmark expects it to call.

---

> > ### Author Response · Authors · 2025-08-08
> > **Response to Q2**
> >
> > We sincerely appreciate your feedback and insights.
> >
> > # Response to Q2.
> > - First, we would like to clarify that Acc-Args is a more challenging metric than F1, as it provides a more comprehensive evaluation of the agent's tool-calling capabilities. Any errors in the tool name, API call format, missing or incorrect parameters are treated as incorrect. **Notably, this metric has also been adopted in previous LLM agent studies [1]**, offering a detailed evaluation in handling both function names and parameters simultaneously.
> > - The reason for adopting this score is that most samples in our dataset involve multiple tools, where the success of certain tools directly impacts others. For example, when checking the weather, the GPS tool must correctly obtain the location before proceeding. Here, we present three examples, including incorrect parameters, incorrect tool names, and incorrect proactive prediction. All of these result in Acc-Args being marked as incorrect. We hope they can help you understand this metric.
> >
> >   ```
> >   "GT":"[{"name":"get_current_datetime", "desc":"Get the current date and time", "params": "None"},{"name": "get_current_gps_coordinates", "desc": "Get the current GPS coordinates of the user", "params": "None"},{"name": "get_city_weather", "desc": "Get the weather for a specified city at a given time.", "params": {"city": "results of get_current_gps_coordinates", "time":"results of get_current_datetime"}}]"
> >
> >   # Incorrect case 1 (incorrect parameters):
> >   "Predictions":"[{"name":"get_current_gps_coordinates","desc":"Get the current GPS coordinates of the user","params": "None"},{"name":"get_city_weather", "desc":"Get the weather for a specified city at a given time.", "params":{"city": "city", "time":"now"}}]"
> >
> >   # Incorrect case 2 (incorrect tool names):
> >   "Predictions":"[{"name":"GPS Location", "params":"None"},{"name":"Weather forecast", "params":{"city":"city", "time":"now"}}]"
> >
> >   # Incorrect case 3 (incorrect proactive prediction):
> >   "Predictions": "None"
> >   ```
> > - Additionally, assigning partial credit for partial matches would improve all methods' metrics but fail to reflect their true performance. In fact, F1 can essentially reflect the scores when the tools are only partially matching the ground-truth set.
> > - Finally, we would like to clarify that the context-aware proactive agent is designed to map rich sensory contexts surrounding humans into proactive predictions with user-intended tool calling. Our results validate the effectiveness of the proposed framework, ContextAgent, while also highlighting room for improvement in accurately mapping contexts to user-intended tool calls. Results also suggest the potential to further enhance performance on this task by continuously improving the capabilities of reasoning models.
> >
> > [1] HammerBench: Fine-Grained Function-Calling Evaluation in Real Mobile Device Scenarios, ACL'25.
> >
> > We would greatly appreciate knowing whether our explanations helped address your concerns. Thank you once again for your time and thoughtful consideration.

---

> > > ### Comment · Reviewer_EoSu · 2025-08-09
> > >
> > > Thank you for the detailed response. It is clear now that this benchmark is far from being saturated and that increasing the number of examples doesn't trivially solve this. The example for Acc-Args is also very informative; my previous understanding was that there may be multiple solutions to a given problem and the agent may be choosing a subset of (independent) tools that provide a user with sufficient information but may not align with the ground truth that provides exhaustive information. The example you provided clearly shows that the ground-truth contains tool calls that are necessary to solve the task (as get_city_weather is dependent on get_current_datetime and get_current_gps_coordinates) and so an exact match makes sense.
> > >
> > > This addresses the remainder of my concerns, I will be raising my score.

---

> ### Author Response · Authors · 2025-08-08
> **Response to Q1**
>
> We sincerely appreciate that our clarifications have addressed your concerns about our dataset, the thresholds we set, and why ProactiveBench is out of scope.
>
> # Response to Q1.
> - First, we would like to clarify that the high performance of Claude Opus 4-ICL is primarily attributed to the model's advanced **intrinsic reasoning capabilities, rather than the high quality of the demonstrations**. As shown in Tab.6 of our paper (same setting as Claude Opus 4), under the same ICL setting with 10 demonstrations, ICL-All on Qwen2.5-7B-Ins achieves only 89.0% Acc-P and 54.3% F1, far below Claude Opus 4's 93.7% Acc-P and 81.8% F1. Under the same base model (Qwen2.5-7B-Ins), ContextAgent outperforms ICL-ALL by +1.9% in Acc-P and +14.6% in F1.
> || Acc-P| MD| FD| RMSE |Precision|Recall|F1| Acc-Args |
> |-|-|-|-|-|-|-|-|-|
> |Claude Opus 4 (ICL-10)|93.7%|4.1%|2.0%|0.86|81.4%|85.6%|81.8%|56.8%|
> |Qwen2.5-7B-Ins (ICL-10)|89.0%|4.0%|7.0%|1.50|59.3%|52.7%|54.3%|20.6%|
> |Qwen2.5-7B-Ins (ContextAgent)|90.9%|2.0%|7.0%|1.17|71.1%|69.9%|68.9%|51.6%|
>
>   Results highlight the potential of reasoning-based LLMs for context-aware proactive agent tasks, suggesting the potential to further enhance performance on this task by continuously improving the capabilities of reasoning models.
>
> &nbsp;
> - Second, obtaining high-quality demos is challenging in real-world applications. Moreover, **simply increasing the number of demonstrations may eventually lead to saturation or marginal effect (limited performance gains while increased computational costs)**. This is why we chose to include 10 demonstrations in the prompt for ICL-based baselines, rather than continuously increasing the number of shots. To validate this, we conduct experiments by increasing the number of examples to 20 (keeping the original 10 and randomly adding new ones). The table below shows that the improvement is limited compared to using only 10 demos. Therefore, **relying solely on a prompt-based solution is not practical for closing the remaining gap.**
> || Acc-P| MD| FD| RMSE |Precision|Recall|F1| Acc-Args |
> |-|-|-|-|-|-|-|-|-|
> |Claude Opus 4 (ZS)|83.0%|7.2%|9.6%|1.27|73.8%|66.0%|68.0%|24.4%|
> |Claude Opus 4 (ICL-10)|93.7%|4.1%|2.0%|0.86|81.4%|85.6%|81.8%|56.8%|
> |Claude Opus 4 (ICL-20)|93.4%|3.5%|3.1%|0.85|82.6%|82.8%|81.2%|57.1%|
> - Existing studies also show that continuously increasing the demos merely brings marginal improvement [1]. For ICL, when an excessive number of demos are included in the prompt, which are concatenated and appended to the query as input, it becomes challenging for an LLM to distinguish and follow the correct demo, especially if its pretraining lacks exposure to such tasks. Continuously increasing the number of demos can also result in significant costs and computational overhead, posing challenges for practical use.
>
> [1] How Many Demonstrations Do You Need for In-context Learning, EMNLP'23.
>
> We would greatly appreciate knowing whether our explanations helped address your concerns. Your support and encouragement are invaluable to us! Thank you once again for your time and thoughtful consideration.

---

> ### Author Response · Authors · 2025-08-09
>
> We are more than happy to address all your concerns and issues. Thank you for the time and effort you have spent reviewing our paper and for sharing your valuable insights, which have helped us enhance its quality. We will add these descriptions to the camera-ready version to further improve the quality of our paper.
>
> Thank you once again for your time and thoughtful consideration. Your support and encouragement are invaluable to us!

---

### Official Review · Reviewer_BDQH · 2025-07-03

**Clarity:** 2
**Significance:** 3
**Originality:** 2
**Rating:** 4
**Confidence:** 3

**Summary:**

This paper introduces ContextAgent for proactive LLM-based agents that leverage rich sensory and persona contexts to provide tool-augmented services without explicit user instructions. By integrating data from wearable devices, ContextAgent autonomously infers user intentions and determines when and how to assist, using external tools accordingly. This paper further propose a benchmark named ContextAgentBench with 1,000 samples across nine daily scenarios to evaluate such agents. Experiments show that ContextAgent significantly outperforms baselines in proactive prediction and tool-calling accuracy.

**Questions:**

See the weaknesses part.

**Ethical Concerns:**

["NO or VERY MINOR ethics concerns only"]

**Limitations:**

yes

**Quality:**

2

**Strengths And Weaknesses:**

**Strengths**
1. The paper presents a compelling approach by leveraging sensory data from wearable devices to enable context-aware, proactive LLM agents. This direction is highly relevant for advancing AI assistants toward more natural, human-centric interactions in real-world environments.
2. The introduction of ContextAgentBench, a comprehensive benchmark designed specifically for evaluating proactive LLM agents in open-world settings, is a valuable contribution. It provides a structured and reproducible way to assess future systems and can foster further research in proactive agent design.

**Weaknesses**
1. While the proposed system is practically useful, the core technical components—such as using VLMs for visual understanding and in-context learning for context extraction—largely rely on existing methodologies. The innovation primarily lies in system integration rather than algorithmic advances, which limits the novelty from a technical standpoint.
2. The paper would benefit from improved clarity and organization. Certain sections are dense and difficult to follow, which may hinder reader comprehension. Enhancing the exposition, especially around key concepts and system design choices, would strengthen the overall readability and impact of the work.

---

> ### Author Rebuttal · Authors · 2025-07-27
>
> Thank you for your valuable feedback on our submission. We sincerely appreciate the reviewer’s recognition of our contribution, which leverages sensory data from wearable devices to develop context-aware, proactive LLM agents, paving the way for more natural and human-centric AI interactions in real-world settings.
>
> # W1. Contribution of this work.
> # Response:
> - First, we would like to clarify that **our work is the first to study: context-aware proactive agents**. Existing studies on LLM agents primarily adopt a reactive paradigm, which requires explicit user instructions to activate their services. In contrast, we first propose a new research problem aimed at leveraging the **massive contexts surrounding humans, which can be captured through various wearable and mobile devices, to enhance the proactivity of LLM agents**.  The paradigm of context-aware proactive agent proposed in this work needs to map sensory contexts into intention understanding and tool calling, rather than relying on direct user language instructions. Effectively leveraging these contexts surrounding humans to develop more intelligent and **proactive** agents is a promising research direction. This advancement will not only benefit the community but also drive the development of advanced, human-centric, and proactive AI assistants.
> - To address this new research problem, we propose a framework called ContextAgent, where we innovatively model this new task by incorporating both sensory contexts and personas for enhanced proactive LLM reasoning. Additionally, we introduce a comprehensive dataset, ContextAgentBench, the first benchmark for context-aware proactive agent tasks, designed to validate this new research problem. Each sample in our dataset includes multimodal sensory perceptions, sensory contexts, personas, reasoning traces (indicating whether and what proactive services are provided), proactive scores, tool chains, and the final responses. To gain a comprehensive understanding of the rationale behind whether proactive services are needed and what proactive services can benefit the user in the current context, we extract reasoning traces distilled from the latest reasoning LLMs. These reasoning traces not only benefit the training of smaller LLMs but also enhance the explainability of this new task. **We believe that our pioneering work on context-aware proactive agents, along with the developed benchmark, will significantly benefit both the AI and human-centric communities.**
>
> We hope this clarification accurately reflects the scope of our work and addresses your concerns.
>
> # W2. Concern about the clarity and organization.
> # Response:
> First, we will enhance the clarity of the main contributions of this work, as well as the motivation and significance of developing context-aware proactive agents. Second, we will enhance the clarity of the design choices for both the ContextAgent architecture and our benchmark, while refining the overall paper to enhance its exposition and readability. Besides, we will add more examples from our constructed dataset in the appendix to help readers better understand our dataset. We will clearly present each component of the dataset, including sensory contexts, personas, reasoning traces, proactive scores, tool usage, and final responses.

---

### Comment · Area_Chair_nUD8 · 2025-08-05
**Please offer your feedback towards author rebuttal**

Dear Reviewers,

Please offer your feedback towards author rebuttal. More discussions are welcome!


Area Chair

---

### Note · Authors · 2025-08-12

We sincerely appreciate your time and effort in reviewing our work. We summarize the rebuttal as follows:
1. Dataset details. Reviewers EoSu and D7kT expressed interest in the examples within our dataset. **We have provided detailed examples in our dataset** and added further explanations, which **have addressed their concerns**.
2. Benchmark saturation.  Reviewer EoSu raised concerns about benchmark saturation. After providing detailed clarifications and additional experiments, the reviewer acknowledged that **all concerns were addressed and raised the score.**
3. Advanced reasoning LLMs and VLMs. We have added new experiments using advanced reasoning LLMs and VLMs. Reviewer HAmH **highly appreciated our experiments on VLM agent and acknowledged that concerns have been addressed, considering raising the score**.
4. Baseline comparison. Reviewers HAmH and D7kT raised concerns regarding the baselines. **We have provided clarifications that the chosen baselines are fair**, noting that for this new task, there are no existing works that can be directly used as baselines. We have utilized Proactive Agent, ICL variants, and added experiments with more SFT-based baselines. Notably, ContextAgent achieves +5.3% Acc-P and 5.2% F1 compared to the best baseline, validating the effectiveness of our design. The clarification **have addressed their concerns**.
5. Special scenarios. Reviewer HAmH is interested in special cases, such as missing sensor data or unavailable tools. We have provided both qualitative and quantitative results, which **have addressed the Reviewer's concerns**.
6. Finally, we would like to provide clarification to the reviewer D7kT's concerns.
   - First, we would like to clarify that **we described the SFT dataset construction process in Sec.4.2 of our original paper. This is consistent with our response in Rebuttal**. We will add the description of the training data size to the camera-ready version.
   - Second, we would like to clarify that **we have provided a detailed analysis of the ICL-based baselines in Appendix E (lines 558–568) in our original paper**.
   - Third, **our clarifications have addressed all the concerns of Reviewers EoSu and HAmH**. They acknowledged our clarification and experiments to help them address their concerns. **These results mainly clarify and enhance the reviewers' understanding, with minimal changes to the original paper. We will improve the clarity of the design and evaluation in our camera-ready version.**

---

### Decision · Program_Chairs · 2025-09-17

**Decision:**

Accept (poster)

**Comment:**

This paper receives three positive scores and one negative score. Overally speaking, the positive points outweigh the negative points. Thus, I recommend accepting this paper.